

# On the R-matrix realization of quantum loop algebras

**Andrii Liashyk[1⋆] and Stanislav Z. Pakuliak[2,3†]**

**1** Skolkovo Institute of Science and Technology, Moscow, Russia,
National Research University Higher School of Economics, Moscow, Russia,
**2** Bogoliubov Laboratory for Theoretical Physics, JINR,
Dubna, Moscow region, Russia,
**3** Landau School of Physics and Research, NRU MIPT,
Dolgoprudny, Moscow region, Russia

⋆ a.liashyk@gmail.com , † pakuliak@jinr.ru

## Abstract

We consider R-matrix realization of the quantum deformations of the loop algebras $\tilde{\mathfrak{g}}$ corresponding to non-exceptional affine Lie algebras of type $\hat{\mathfrak{g}} = A_{N-1}^{(1)}, B_n^{(1)}, C_n^{(1)}, D_n^{(1)}, A_{N-1}^{(2)}$. For each $U_q(\tilde{\mathfrak{g}})$ we investigate the commutation relations between Gauss coordinates of the fundamental L-operators using embedding of the smaller algebra into bigger one. The new realization of these algebras in terms of the currents is given. The relations between all off-diagonal Gauss coordinates and certain projections from the ordered products of the currents are presented. These relations are important in applications to the quantum integrable models.

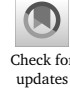

# 1   Introduction

Classification of the solutions to the quantum Yang-Baxter equation for the case of non-exceptional quantum affine Lie algebras was found in the pioneering paper [1].

Let $\mathfrak{g}$ be one of the Lie algebras $\mathfrak{sl}_N$, $\mathfrak{o}_{2n+1}$, $\mathfrak{sp}_{2n}$ or $\mathfrak{o}_{2n}$ corresponding to the series of the classical Lie algebras $A_{N-1}$, $B_n$, $C_n$ and $D_n$ respectively. Let $\widehat{\mathfrak{g}}$ be one of non-exceptional affine Lie algebras $A_{N-1}^{(1)}$, $B_n^{(1)}$, $C_n^{(1)}$, $D_n^{(1)}$ and $A_{N-1}^{(2)}$. By $\tilde{\mathfrak{g}}$ we denote the loop algebra which is the affine algebra $\widehat{\mathfrak{g}}$ with zero central charge. To save notations we will use the same names for the different loop algebras $\tilde{\mathfrak{g}}$ as for the affine algebras $\widehat{\mathfrak{g}}$.

Let $q \in \mathbb{C}$ be an arbitrary complex number not equal to zero or root of unity. In this paper we consider quantum deformation $U_q(\tilde{\mathfrak{g}})$ [2] of the universal enveloping algebra $U(\tilde{\mathfrak{g}})$ which we call the quantum loop algebra. One may think about $U_q(\tilde{\mathfrak{g}})$ as the corresponding quantum affine algebra $U_q(\widehat{\mathfrak{g}})$ with zero central charge.

Algebra $U_q(\tilde{\mathfrak{g}})$ has several descriptions. It can be formulated in terms of the finite number of Chevalley generators or countable set of Cartan-Weyl generators. Latter generators can be gathered into finite number of the generating series and the commutation relations between whole set of the Cartan-Weyl generators can be realized as finite number of the formal series relations between these generating series.

For the applications to the quantum integrable models, the second description of $U_q(\tilde{\mathfrak{g}})$ is more suitable since generating series of the Cartan-Weyl generators can be identified with Gauss coordinates of the fundamental L-operators, which satisfy the same RLL-type commutation relations as quantum monodromies of the integrable systems do. It opens a possibility to construct off-shell Bethe vectors for these integrable models in terms of Cartan-Weyl generators of the algebra $U_q(\tilde{\mathfrak{g}})$ [3].

Realization of the algebra $U_q(\tilde{\mathfrak{g}})$ in terms of Cartan-Weyl generators has in turn two faces. One is given by the quadratic RLL-type commutation relations for the fundamental L-operators

defined by the solution of the quantum Yang-Baxter equation [1]. This construction was first proposed in the paper [4]. On the other hand the algebra $U_q(\tilde{\mathfrak{g}})$ can be realized in terms of so called *currents* [5]. For the case $U_q(\widehat{\mathfrak{gl}}_N)$ an isomorphism between these two descriptions was found in [6] . Recent papers [7, 8] prove similar isomorphisms for the algebras $U_q(B_n^{(1)})$, $U_q(C_n^{(1)})$ and $U_q(D_n^{(1)})$. In our investigation we extend these results to the case of $U_q(A_{N-1}^{(2)})$. Key observation is the fact that R-matrix associated with the algebras $U_q(B_n^{(1)})$, $U_q(C_n^{(1)})$, $U_q(D_n^{(1)})$, $U_q(A_{N-1}^{(2)})$ has the same structure for all these algebras. The differences are accumulated in one parameter $\xi$ (see (2.1)).

In [1] one more solution to the quantum Yang-Baxter equation was found. It corresponds to the affine algebra $D_n^{(2)}$. This solution has more complicated structure than R-matrices for above mentioned algebras. We will describe the corresponding quantum loop algebra $U_q(D_n^{(2)})$ in our future publications.

The paper is composed as follows. In section 2 quantum R-matrix for the algebra $U_q(\tilde{\mathfrak{g}})$ is defined together with its properties. Section 3 is devoted to definition of the algebra $U_q(\tilde{\mathfrak{g}})$ and description of its central elements and automorphism. Gauss coordinates of the fundamental L-operators are introduced in section 4. Here we discuss normal ordering of subalgebras in $U_q(\tilde{\mathfrak{g}})$ induced by the cyclic ordering of the Cartan-Weyl generators in the quantum affine algebras. Section 5 contains the theorem which describes embedding of the smaller rank algebra $U_q(\tilde{\mathfrak{g}})$ into the bigger one. This embedding is described on the level of matrix entries of the fundamental L-operators and in terms of the Gauss coordinates. Section 6 describes new realization of the algebra $U_q(\tilde{\mathfrak{g}})$ in terms of the currents. In section 7 so called composed currents are introduced which belong to certain completion of $U_q(\tilde{\mathfrak{g}})$ and related to off-diagonal Gauss coordinates of the fundamental L-operators. It was shown in [9, 10] that analytical properties of the composed currents and the commutation relations between them are equivalent to the Serre relations between simple root currents. Proofs of auxiliary Propositions and Lemmas are gathered in four Appendices.

# 2   R-matrix for $U_q(\tilde{\mathfrak{g}})$

Let $N$ be dimension of the fundamental vector representation of the algebra $\tilde{\mathfrak{g}}$ in $\mathbb{C}^N$. Let $\mathsf{e}_{ij}$ be an $N \times N$ matrix unit $(\mathsf{e}_{ij})_{k,l} = \delta_{ik}\delta_{jl}$ for $1 \leq i,j,k,l \leq N$ and

$$i' = N + 1 - i, \quad 1 \leq i \leq N.$$

To describe quantum R-matrix associated with the algebra $U_q(\tilde{\mathfrak{g}})$ [1, 7, 8] we define parameter $\xi$ and dimension $N$ of the fundamental vector representation of the algebra $\tilde{\mathfrak{g}}$ given in the table

| $\tilde{\mathfrak{g}}$ | $A_{N-1}^{(1)}$ | $B_n^{(1)}$ | $C_n^{(1)}$ | $D_n^{(1)}$ | $A_{2n}^{(2)}$ | $A_{2n-1}^{(2)}$ |
|---|---|---|---|---|---|---|
| $N$ | $N$ | $2n+1$ | $2n$ | $2n$ | $2n+1$ | $2n$ |
| $\xi$ | $q^{-N}$ | $q^{1-2n}$ | $q^{-2-2n}$ | $q^{2-2n}$ | $-q^{-1-2n}$ | $-q^{-2n}$ |

(2.1)

Define also the sign function

$$\mathrm{sign}(\ell) = \begin{cases} +1, & \ell \geq 0 \\ -1, & \ell < 0 \end{cases}$$

and a set of integers $\varepsilon_i$, $i = 1, \ldots, N$

$$\varepsilon_i = \begin{cases} \mathrm{sign}(n-i), & \text{for} \quad \tilde{\mathfrak{g}} = C_n^{(1)}, \\ 1, & \text{for} \quad \text{all other cases}. \end{cases}$$

For $\tilde{\mathfrak{g}} = A_{N-1}^{(1)}, B_n^{(1)}, C_n^{(1)}, D_n^{(1)}$ and $A_{N-1}^{(2)}$ we need the map $\bar{\imath}$ for $i = 1, \ldots, N$

$$
\bar{\imath} = \begin{cases}
\left(\frac{N}{2} - \frac{1}{2}, \frac{N}{2} - \frac{3}{2}, \ldots, -\frac{N}{2} + \frac{3}{2}, -\frac{N}{2} + \frac{1}{2}\right), & \text{for} \quad \tilde{\mathfrak{g}} = A_{N-1}^{(1)}, \\
\left(n - \frac{1}{2}, \ldots, \frac{3}{2}, \frac{1}{2}, 0, -\frac{1}{2}, -\frac{3}{2}, \ldots, -n + \frac{1}{2}\right), & \text{for} \quad \tilde{\mathfrak{g}} = B_n^{(1)}, A_{2n}^{(2)}, \\
(n, n-1, \ldots, 1, -1, \ldots, -n), & \text{for} \quad \tilde{\mathfrak{g}} = C_n^{(1)}, \\
(n-1, \ldots, 1, 0, 0, -1, \ldots, -n+1), & \text{for} \quad \tilde{\mathfrak{g}} = D_n^{(1)}, A_{2n-1}^{(2)}.
\end{cases}
\tag{2.2}
$$

Note that for any $\tilde{\mathfrak{g}}$ we have

$$
\bar{\imath} + \bar{\imath}' = 0, \quad i = 1, \ldots, N. \tag{2.3}
$$

We introduce functions

$$
f(u, v) = \frac{qu - q^{-1}v}{u - v}, \quad g(u, v) = \frac{(q - q^{-1})u}{u - v}, \quad \tilde{g}(u, v) = \frac{(q - q^{-1})v}{u - v}
$$

of the arbitrary complex numbers $u$ and $v$, which we call the spectral parameters.

Define matrices $\mathbb{P}(u, v)$ and $\mathbb{Q}(u, v)$ acting in the tensor product $\mathbb{C}^N \otimes \mathbb{C}^N$

$$
\mathbb{P}(u, v) = \sum_{1 \le i, j \le N} \mathsf{p}_{ij}(u, v)\, \mathsf{e}_{ij} \otimes \mathsf{e}_{ji}, \tag{2.4}
$$

$$
\mathbb{Q}(u, v) = \sum_{1 \le i, j \le N} \mathsf{q}_{ij}(u, v)\, \mathsf{e}_{i'j'} \otimes \mathsf{e}_{ij}, \tag{2.5}
$$

where rational functions $\mathsf{p}_{ij}(u, v)$ and $\mathsf{q}_{ij}(u, v)$ are defined as follows

$$
\mathsf{p}_{ij}(u, v) = \begin{cases}
f(u, v) - 1, & i = j, \\
g(u, v), & i < j, \\
\tilde{g}(u, v), & i > j,
\end{cases}
$$

$$
\mathsf{q}_{ij}(u, v) = \varepsilon_i \varepsilon_j\, q^{\bar{\imath} - \bar{\jmath}} \begin{cases}
f(v\xi, u) - 1, & i = j, \quad i \ne i', \\
f(v\xi, u) - 1 - \alpha_q, & i = j, \quad i = i', \\
g(v\xi, u), & i < j, \\
\tilde{g}(v\xi, u), & i > j,
\end{cases}
\tag{2.6}
$$

and

$$
\alpha_q = (q^{1/2} - q^{-1/2})^2. 
$$

One can check that functions (2.6) have a property

$$
q_{ij}(u, v) = q_{j'i'}(u, v). \tag{2.7}
$$

Let

$$
\mathbb{I} = \sum_{i=1}^{N} \mathsf{e}_{ii}
$$

be identity matrix in $\mathbb{C}^N$.

**Definition 2.1.** *Quantum trigonometric R-matrix acting in the tensor product of two fundamental vector representations of $\tilde{\mathfrak{g}}$ [1] for the algebra $\tilde{\mathfrak{g}} = A_{N-1}^{(1)}$ is*

$$
\mathbb{R}(u, v) = \mathbb{I} \otimes \mathbb{I} + \mathbb{P}(u, v) \tag{2.8}
$$

*and for the algebras $\tilde{\mathfrak{g}} = B_n^{(1)}, C_n^{(1)}, D_n^{(1)}$ and $A_{N-1}^{(2)}$ is*

$$
\mathsf{R}(u, v) = \mathbb{R}(u, v) + \mathbb{Q}(u, v) = \mathbb{I} \otimes \mathbb{I} + \mathbb{P}(u, v) + \mathbb{Q}(u, v). \tag{2.9}
$$

For any $X \in \text{End}(\mathbb{C}^N)$ transposed matrix $X^{\text{t}}$ is

$$(X^{\text{t}})_{i,j} = \varepsilon_i \, \varepsilon_j \, X_{j',i'} \, . \tag{2.10}$$

Let D be a diagonal matrix

$$D = \text{diag}(q^{\bar{1}}, q^{\bar{2}}, \dots, q^{\bar{N}}),$$

where $\bar{\imath}$ for $i = 1, \dots, N$ are given by (2.2).

Let P be permutation operator ($\text{P}^2 = \mathbb{I}$) in $\mathbb{C}^N \otimes \mathbb{C}^N$ and Q be projector ($\text{Q}^2 = N\text{Q}$) onto a one-dimensional subspace in $\mathbb{C}^N \otimes \mathbb{C}^N$

$$\text{P} = \sum_{1 \leq i,j \leq N} \text{e}_{ij} \otimes \text{e}_{ji} \, , \quad \text{Q} = \sum_{1 \leq i,j \leq N} \varepsilon_i \varepsilon_j \text{e}_{i'j'} \otimes \text{e}_{ij} = \text{P}^{\text{t}_1} = \text{P}^{\text{t}_2} \, .$$

Trigonometric $\tilde{\mathfrak{g}}$-invariant R-matrix given by (2.8) and (2.9) possesses following properties.

- *Scaling invariance*

$$\text{R}(\beta u, \beta v) = \text{R}(u, v), \tag{2.11}$$

  for any complex parameter $\beta$ which is not equal to zero.

- *Transposition symmetry*

$$\text{R}_{12}(u, v)^{\text{t}_1 \text{t}_2} = \text{R}_{12}(u, v) \, . \tag{2.12}$$

- *Twist symmetry*

$$K_1 \, K_2 \, \text{R}_{12}(u, v) = \text{R}_{12}(u, v) \, K_1 \, K_2 \, , \tag{2.13}$$

  where $K$ is $N \times N$ $\mathbb{C}$-valued matrix such that $KK^{\text{t}} = \mathbb{I}$, $K_1 = K \otimes \mathbb{I}$ and $K_2 = \mathbb{I} \otimes K$. The equality (2.13) is valid for $K = D$ since $D^{\text{t}} = D^{-1}$ due to (2.3).

- *Yang-Baxter equation*

$$\text{R}_{12}(u, v) \cdot \text{R}_{13}(u, w) \cdot \text{R}_{23}(v, w) = \text{R}_{23}(v, w) \cdot \text{R}_{13}(u, w) \cdot \text{R}_{12}(u, v), \tag{2.14}$$

  where subscripts of R-matrices mean the indices of the spaces $\mathbb{C}^N$ where it acts nontrivially.

- *Unitarity*

$$\text{R}_{12}(u, v) \cdot \text{R}_{21}(v, u) = f(u, v) f(v, u) \, \mathbb{I} \otimes \mathbb{I}, \tag{2.15}$$

  where $\text{R}_{21}(u, v) = \text{P}_{12} \, \text{R}_{12}(u, v) \, \text{P}_{12}$.

- *Crossing type symmetries*

$$\text{D}_1^2 \, \mathbb{R}_{12}(v\xi^2, u)^{\text{t}_1} \, \text{D}_1^{-2} \, \mathbb{R}_{21}(u, v)^{\text{t}_1} = \mathbb{I} \otimes \mathbb{I}, \tag{2.16}$$

  for $\tilde{\mathfrak{g}} = A_{N-1}^{(1)}$ with $\xi = q^{-N}$ and

$$\text{D}_1 \, \text{R}_{12}(v\xi, u)^{\text{t}_1} \, \text{D}_1^{-1} \, \text{R}_{12}(v, u), = f(u, v) f(v, u) \, \mathbb{I} \otimes \mathbb{I} \tag{2.17}$$

  for $\tilde{\mathfrak{g}} = B_n^{(1)}, C_n^{(1)}, D_n^{(1)}, A_{N-1}^{(2)}$. Crossing relation (2.17) follows from the presentation of the matrix $\mathbb{Q}(u, v)$ given by (2.5) in the form

$$\mathbb{Q}(u, v) = \text{D}_2 \, \text{P}_{12} \, \mathbb{P}_{12}(v\xi, u)^{\text{t}_1} \, \text{P}_{12} \, \text{D}_2^{-1} - \alpha_q \delta_{N,\text{odd}} \, \text{e}_{n+1,n+1} \otimes \text{e}_{n+1,n+1},$$

  where $\delta_{N,\text{odd}} = 1$ for $N = 2n + 1$ and 0 for $N = 2n$. This relation implies

$$\text{R}_{12}(u, v) = \text{D}_2 \, \text{P}_{12} \, \text{R}_{12}(v\xi, u)^{\text{t}_1} \, \text{P}_{12} \, \text{D}_2^{-1}$$

  which is equivalent to (2.17) due to (2.15). Crossing symmetry (2.17) for R-matrix (2.9) yields the relation similar to (2.16)

$$\text{D}_1^2 \, \text{R}_{12}(v\xi^2, u)^{\text{t}_1} \, \text{D}_1^{-2} \, \text{R}_{21}(u, v)^{\text{t}_1} = f(u, v\xi) f(v\xi, u) \, \mathbb{I} \otimes \mathbb{I} \tag{2.18}$$

- *Pole structure*
  R-matrix (2.8) for $\tilde{\mathfrak{g}} = A^{(1)}_{N-1}$ has a simple pole at $u = v$

$$\frac{(u-v)}{u(q-q^{-1})} \, \mathbb{R}_{12}(u,v)\bigg|_{u=v} = \mathsf{P}_{12}, \tag{2.19}$$

while R-matrix (2.9) for $\tilde{\mathfrak{g}} = B^{(1)}_n, C^{(1)}_n, D^{(1)}_n, A^{(2)}_{N-1}$ has two simple poles at $u = v$ and $u = v\xi$ with residues

$$\frac{(u-v)}{u(q-q^{-1})} \, \mathsf{R}_{12}(u,v)\bigg|_{u=v} = \mathsf{P}_{12}, \qquad \frac{(v\xi-u)}{u(q-q^{-1})} \, \mathsf{R}_{12}(u,v)\bigg|_{u=v\xi} = \mathsf{D}_1^{-1}\,\mathsf{P}_{12}^{\mathsf{t}_1}\,\mathsf{D}_1. \tag{2.20}$$

This pole structure and crossing relations (2.16) and (2.18) allow to get for both R-matrices (2.8) and (2.9)

$$x(u,v\xi^2)^{-1} \, \frac{v\xi^2-u}{u(q-q^{-1})}\Big(\mathsf{R}_{12}(u,v)^{\mathsf{t}_1}\Big)^{-1}\bigg|_{u=v\xi^2} = \mathsf{D}_1^{-2}\,\mathsf{P}_{12}^{\mathsf{t}_1}\,\mathsf{D}_1^2, \tag{2.21}$$

where rational function $x(u,v)$ is

$$x(u,v) = \begin{cases} 1, & \text{for} \quad \tilde{\mathfrak{g}} = A^{(1)}_{N-1}, \\ f(u\xi,v)^{-1}f(v,u\xi)^{-1} & \text{for all other} \quad \tilde{\mathfrak{g}}. \end{cases} \tag{2.22}$$

- *Scaling limit*
  In the scaling limit $\epsilon \to 0$ and $u \to e^{\epsilon u}$, $v \to e^{\epsilon v}$, $q \to e^{\epsilon c/2}$, $\xi \to e^{-\epsilon c\kappa}$ trigonometric $R$-matrix (2.9) goes into rational $\mathfrak{g}$-invariant $R$-matrix

$$\mathsf{R}(u,v) = \mathbb{I} \otimes \mathbb{I} + \frac{c}{u-v}\,\mathsf{P} - \frac{c}{u-v+c\kappa}\,\mathsf{Q}, \tag{2.23}$$

for the algebras $\tilde{\mathfrak{g}} = B^{(1)}_n, C^{(1)}_n, D^{(1)}_n$ and into rational $\mathfrak{gl}_N$-invariant the R-matrix

$$\mathbb{R}(u,v) = \mathbb{I} \otimes \mathbb{I} + \frac{c}{u-v}\,\mathsf{P},$$

for the algebras $\tilde{\mathfrak{g}} = A^{(1)}_{N-1}, A^{(2)}_{N-1}$.

Quantum R-matrix (2.23) appeared in investigation of the classical series Yangians and their doubles in [11–13].

## 3 R-matrix formulation of the algebra $U_q(\tilde{\mathfrak{g}})$

The algebra $U_q(\tilde{\mathfrak{g}})$ over $\mathbb{C}(q)$ (over $\mathbb{C}(q^{1/2})$ for $\tilde{\mathfrak{g}} = B^{(1)}_n$ and $A^{(2)}_{2n}$) is generated by the elements $\mathsf{L}^{\pm}_{i,j}[\pm m]$, $1 \le i, j \le N$, $m \in \mathbb{Z}_+$ such that

$$\mathsf{L}^+_{j,i}[0] = \mathsf{L}^-_{i,j}[0] = 0, \quad i < j, \quad \mathsf{L}^+_{i,i}[0]\mathsf{L}^-_{i,i}[0] = \mathsf{L}^-_{i,i}[0]\mathsf{L}^+_{i,i}[0] = 1. \tag{3.24}$$

There are also additional relations for the operators $\mathsf{L}^{\pm}_{i,j}[\pm m]$ which are due to existence of the central elements in $U_q(\tilde{\mathfrak{g}})$ described in section 3.1.

The generators of the algebra $U_q(\tilde{\mathfrak{g}})$ can be gathered into formal series

$$\mathsf{L}^{\pm}_{i,j}(u) = \sum_{m=0}^{\infty} \mathsf{L}^{\pm}_{i,j}[\pm m]u^{\mp m} \tag{3.25}$$

and combined in the matrices

$$\text{L}^{\pm}(u) = \sum_{i,j=1}^{N} \text{e}_{ij} \otimes \text{L}^{\pm}_{i,j}(u) \in \text{End}(\mathbb{C}^N) \otimes U_q(\tilde{\mathfrak{g}})[[u,u^{-1}]], \tag{3.26}$$

which we call L-operators[1]. The commutation relations in the algebra $U_q(\tilde{\mathfrak{g}})$ are given by the standard RLL commutation relations in $(\mathbb{C}^N)^{\otimes 2} \otimes U_q(\tilde{\mathfrak{g}})[[u,u^{-1}]]$

$$\text{R}(u,v) \cdot (\text{L}^{\mu}(u) \otimes \mathbb{I}) \cdot (\mathbb{I} \otimes \text{L}^{\rho}(v)) = (\mathbb{I} \otimes \text{L}^{\rho}(v)) \cdot (\text{L}^{\mu}(u) \otimes \mathbb{I}) \cdot \text{R}(u,v), \tag{3.27}$$

where $\mu, \rho = \pm$ and rational functions entering R-matrices (2.8) and (2.9) should be understood as series over $v/u$ for $\mu = +$, $\rho = -$ and as series over $u/v$ for $\mu = -$, $\rho = +$. For $\mu = \rho$ these rational functions can be either series over the ratio $v/u$ or the ratio $u/v$.

The commutation relations in the algebra $U_q(\tilde{\mathfrak{g}})$ may be written in terms of matrix entries (3.25). Using explicit expression (2.4) and (2.5) one gets

$$\begin{aligned}
[\text{L}^{\mu}_{i,j}(u), \text{L}^{\rho}_{k,l}(v)] &= \text{p}_{lj}(u,v)\, \text{L}^{\rho}_{k,j}(v)\text{L}^{\mu}_{i,l}(u) - \text{p}_{ik}(u,v)\, \text{L}^{\mu}_{k,j}(u)\text{L}^{\rho}_{i,l}(v) \\
&\quad + \sum_{p=1}^{N}\left(\delta_{l,j'}\, \text{q}_{pl}(u,v)\, \text{L}^{\rho}_{k,p}(v)\text{L}^{\mu}_{i,p'}(u) - \delta_{i,k'}\, \text{q}_{kp}(u,v)\, \text{L}^{\mu}_{p',j}(u)\text{L}^{\rho}_{p,l}(v)\right).
\end{aligned} \tag{3.28}$$

The sum in the last line of the commutation relations (3.28) is absent for the algebra $\tilde{\mathfrak{g}} = A^{(1)}_{N-1}$. It follows from the commutation relations (3.27) or (3.28) that modes $\text{L}^{+}_{i,j}[m]$ and $\text{L}^{-}_{i,j}[-m]$, $m \geq 0$ form Borel subalgebras $U^{\pm}_q(\tilde{\mathfrak{g}}) \subset U_q(\tilde{\mathfrak{g}})$.

**Remark 3.1.** One can check that the restrictions to the zero mode generators (3.24) are consistent with the commutation relations (3.28). Indeed, taking the limit $u \to \infty$ in (3.28) with $\mu = +$ and using the expansion (3.25), one gets

$$\begin{aligned}
&q^{\delta_{ik}}\text{L}^{+}_{i,j}[0]\,\text{L}^{\rho}_{k,l}(v) - q^{\delta_{jl}}\text{L}^{\rho}_{k,l}(v)\,\text{L}^{+}_{i,j}[0] \\
&= (q - q^{-1})\left(\delta_{l<j}\, \text{L}^{\rho}_{k,j}(v)\text{L}^{+}_{i,l}[0] - \delta_{i<k}\, \text{L}^{+}_{k,j}[0]\text{L}^{\rho}_{i,l}(v)\right) \\
&\quad + \sum_{p=1}^{N}\left(\delta_{l,j'}\text{q}_{pl}\text{L}^{\rho}_{k,p}(v)\text{L}^{+}_{i,p'}[0] - \delta_{i,k'}\text{q}_{kp}\text{L}^{+}_{p',j}[0]\text{L}^{\rho}_{p,l}(v)\right),
\end{aligned} \tag{3.29}$$

where $\delta_{i<j} = 1$ if $i < j$ and 0 otherwise and

$$\text{q}_{ij} = \varepsilon_i \varepsilon_j\, q^{\bar{i}-\bar{j}}\begin{cases} q^{-1} - 1, & i = j, \quad i \neq i', \\ 1 - q, & i = j, \quad i = i', \\ 0, & i < j, \\ -q^{\bar{i}-\bar{j}}(q - q^{-1}), & i > j. \end{cases}$$

Now, if one supposes that $i > j$ and applying (3.24) for the zero mode operators $\text{L}^{+}_{i,j}[0]$, the l.h.s. of (3.29) vanishes identically. Due to the coefficients $\delta_{l<j}$ and $\delta_{i<k}$ in the second line of (3.29) and the combinations $\text{q}_{pj'}\text{L}^{+}_{i,p'}[0]$, $\text{q}_{i'p}\text{L}^{+}_{p',j}[0]$ in the third line of this equality, the r.h.s. also vanishes for the same reason.

Analogously, one can check that the restriction that zero mode operators $\text{L}^{-}_{i,j}[0]$ vanishes for $i < j$ is consistent with the series expansion (3.25) in $u$ of the $L$-operator $\text{L}^{-}(u)$. In that case, the zero modes occur in the limit $u \to 0$, which changes the exchange relations (3.29) and makes everything consistent again. Finally, one can also verify that the limit $v \to \infty$ in (3.28) for $\text{L}^{+}_{k,l}(v)$ and the limit $v \to 0$ for $\text{L}^{-}_{k,l}(v)$ leads to the same conclusions.

---

[1]Further on we will skip the sign $\otimes$ of the tensor product in (3.26) and write simply $\text{L}^{\pm}(u) = \sum_{i,j} \text{e}_{ij}\text{L}^{\pm}_{i,j}(u)$.

## 3.1 Central elements in $U_q(\tilde{\mathfrak{g}})$

Due to the commutation relations (3.27) algebra $U_q(\tilde{\mathfrak{g}})$ has central elements which are given by the following

**Proposition 3.1.** *For any* $\tilde{\mathfrak{g}} = A_{N-1}^{(1)}, A_{N-1}^{(2)}, B_n^{(1)}, C_n^{(1)}$ *and* $D_n^{(1)}$ *the algebra* $U_q(\tilde{\mathfrak{g}})$ *has the central elements* $Z^{\pm}(v) \in U_q^{\pm}(\tilde{\mathfrak{g}})$

$$Z^{\pm}(v)\,\mathbb{I} = \mathrm{D}^2\,\mathrm{L}^{\pm}(v\xi^2)^{\mathrm{t}}\,\mathrm{D}^{-2}\left(\mathrm{L}^{\pm}(v)^{-1}\right)^{\mathrm{t}} = \left(\mathrm{L}^{\pm}(v)^{-1}\right)^{\mathrm{t}}\,\mathrm{D}^2\,\mathrm{L}^{\pm}(v\xi^2)^{\mathrm{t}}\,\mathrm{D}^{-2}\,, \tag{3.30}$$

*where parameter* $\xi$ *is given by the table (2.1). Equation (3.30) means that products of the matrices*

$$\mathrm{D}^2\,\mathrm{L}^{\pm}(v\xi^2)^{\mathrm{t}}\,\mathrm{D}^{-2}\left(\mathrm{L}^{\pm}(v)^{-1}\right)^{\mathrm{t}} \quad and \quad \left(\mathrm{L}^{\pm}(v)^{-1}\right)^{\mathrm{t}}\,\mathrm{D}^2\,\mathrm{L}^{\pm}(v\xi^2)^{\mathrm{t}}\,\mathrm{D}^{-2}$$

*are equal and proportional to the unit matrix* $\mathbb{I}$*. The proportionality coefficients are the central elements.*

*Proof.* To find central elements (3.30) one can transform the commutation relations for the fundamental L-operators (3.27) to the form[2]

$$\left(\mathrm{R}_{12}(u,v)^{\mathrm{t}_1}\right)^{-1}\mathrm{L}^{(2)}(v)^{-1}\,\mathrm{L}^{(1)}(u)^{\mathrm{t}_1} = \mathrm{L}^{(1)}(u)^{\mathrm{t}_1}\,\mathrm{L}^{(2)}(v)^{-1}\left(\mathrm{R}_{12}(u,v)^{\mathrm{t}_1}\right)^{-1}\,,$$

where standard notations

$$\mathrm{L}^{(1)}(u) = \mathrm{L}(u)\otimes\mathbb{I}\,, \qquad \mathrm{L}^{(2)}(u) = \mathbb{I}\otimes\mathrm{L}(u)$$

are used. Taking the residue at the point $u = v\xi^2$ in this equation and using (2.21) one gets

$$\mathrm{D}_1^{-2}\,\mathrm{P}_{12}^{\mathrm{t}_1}\,\mathrm{D}_1^2\,\mathrm{L}^{(2)}(v)^{-1}\,\mathrm{L}^{(1)}(v\xi^2)^{\mathrm{t}_1} = \mathrm{L}^{(1)}(v\xi^2)^{\mathrm{t}_1}\,\mathrm{L}^{(2)}(v)^{-1}\,\mathrm{D}_1^{-2}\,\mathrm{P}_{12}^{\mathrm{t}_1}\,\mathrm{D}_1^2$$

or

$$\left(\mathrm{L}^{(1)}(v)^{-1}\right)^{\mathrm{t}_1}\,\mathrm{D}_1^2\,\mathrm{L}^{(1)}(v\xi^2)^{\mathrm{t}_1}\,\mathrm{D}_1^{-2} = \mathrm{D}_2^2\,\mathrm{L}^{(2)}(v\xi^2)^{\mathrm{t}_2}\,\mathrm{D}_2^{-2}\left(\mathrm{L}^{(2)}(v)^{-1}\right)^{\mathrm{t}_2}\,,$$

which proves equality in (3.30). To prove centrality of the elements $Z^{\pm}(v)$ we consider the chain of equalities

$$\begin{aligned}
Z(u)\mathbb{I}_1\,\mathrm{L}^{(2)}(v) &= \mathrm{D}_1^2\,\mathrm{L}^{(1)}(u\xi^2)^{\mathrm{t}_1}\,\mathrm{D}_1^{-2}\left(\mathrm{L}^{(1)}(u)^{-1}\right)^{\mathrm{t}_1}\mathrm{L}^{(2)}(v) \\
&= \mathrm{D}_1^2\,\mathrm{L}^{(1)}(u\xi^2)^{\mathrm{t}_1}\,\mathrm{D}_1^{-2}\left(\mathrm{R}_{21}(v,u)^{\mathrm{t}_1}\right)^{-1}\mathrm{L}^{(2)}(v)\left(\mathrm{L}^{(1)}(u)^{-1}\right)^{\mathrm{t}_1}\mathrm{R}_{21}(v,u)^{\mathrm{t}_1} \\
&= x(u,v)\mathrm{D}_1^2\,\mathrm{L}^{(1)}(u\xi^2)^{\mathrm{t}_1}\mathrm{R}_{12}(u\xi^2,v)^{\mathrm{t}_1}\mathrm{L}^{(2)}(v)\,\mathrm{D}_1^{-2}\left(\mathrm{L}^{(1)}(u)^{-1}\right)^{\mathrm{t}_1}\mathrm{R}_{21}(v,u)^{\mathrm{t}_1} \\
&= x(u,v)\mathrm{L}^{(2)}(v)\,\mathrm{D}_1^2\,\mathrm{R}_{12}(u\xi^2,v)^{\mathrm{t}_1}\mathrm{L}^{(1)}(u\xi^2)^{\mathrm{t}_1}\mathrm{D}_1^{-2}\left(\mathrm{L}^{(1)}(u)^{-1}\right)^{\mathrm{t}_1}\mathrm{R}_{21}(v,u)^{\mathrm{t}_1} \\
&= \mathrm{L}^{(2)}(v)\left(\mathrm{R}_{21}(v,u)^{\mathrm{t}_1}\right)^{-1}Z(u)\mathbb{I}_1\,\mathrm{R}_{21}(v,u)^{\mathrm{t}_1} = \mathrm{L}^{(2)}(v)\,Z(u)\mathbb{I}_1\,,
\end{aligned}$$

where $x(u,v)$ is defined by (2.22). For these calculations one has to use RLL commutation relations and equalities (2.16) and (2.18) for R-matrices (2.8) and (2.9). $\qquad\square$

---

[2]In what follows we will sometimes skip superscripts of L-operators. If these superscripts is not explicitly mentioned it means that the corresponding relation is valid for both values $\pm$.

**Remark 3.2.** Existence of the central element $Z(u)$ for the Yangian $Y(\mathfrak{gl}_N)$ was mentioned in [14]. In this paper a *quantum Liouville formula* for the Yangian was considered. Analogous relation in the case of the algebra $U_q(A_{N-1}^{(1)})$ takes the form

$$Z^{\pm}(v) = \prod_{s=1}^{N} \frac{k_s^{\pm}(vq^{-2s})}{k_s^{\pm}(vq^{-2(s-1)})} = \frac{\text{q-det}\left(L^{\pm}(vq^{-2})\right)}{\text{q-det}\left(L^{\pm}(v)\right)} \tag{3.31}$$

and can be proved in the same way as in the Yangian case [15]. In (3.31) $k_\ell^{\pm}(v)$ are diagonal Gauss coordinates introduced by (4.37).

We set the central elements $Z^{\pm}(u)$ equal to 1 in the algebra $U_q(\tilde{\mathfrak{g}})$. We denote by $\tilde{U}_q(A_{N-1}^{(1)})$ the algebra defined by $U_q(\mathfrak{gl}_N)$-invariant R-matrix (2.8) without any restrictions to these central elements.

The pole structure of R-matrix for $\tilde{\mathfrak{g}} = B_n^{(1)}, C_n^{(1)}, D_n^{(1)}, A_{N-1}^{(2)}$ given by (2.20) yields other central elements in the corresponding algebras $U_q(\tilde{\mathfrak{g}})$. We have following

**Proposition 3.2.** *There are central elements $z^{\pm}(v) \in U_q^{\pm}(\tilde{\mathfrak{g}})$ for $\tilde{\mathfrak{g}} = B_n^{(1)}, C_n^{(1)}, D_n^{(1)}$ and $A_{N-1}^{(2)}$ given by the equalities*

$$z^{\pm}(v)\, \mathbb{I} = D\, L^{\pm}(v\xi)^{\text{t}}\, D^{-1}\, L^{\pm}(v) = L^{\pm}(v)\, D\, L^{\pm}(v\xi)^{\text{t}}\, D^{-1}. \tag{3.32}$$

*Again,* (3.32) *means that products of the matrices*

$$D\, L^{\pm}(v\xi)^{\text{t}}\, D^{-1}\, L^{\pm}(v) \quad and \quad L^{\pm}(v)\, D\, L^{\pm}(v\xi)^{\text{t}}\, D^{-1}$$

*are proportional to the unity operator $\mathbb{I}$ and the proportionality coefficients are central elements. They are related to $Z^{\pm}(v)$ by the relations*

$$Z^{\pm}(v) = z^{\pm}(v\xi)\, z^{\pm}(v)^{-1}. \tag{3.33}$$

*Proof.* Calculating residue at $u = v\xi$ in the commutation relation (3.27) one gets

$$D_1^{-1}\, P_{12}^{\text{t}_1}\, D_1\, L^{(1)}(v\xi)\, L^{(2)}(v) = L^{(2)}(v)\, L^{(1)}(v\xi)\, D_1^{-1}\, P_{12}^{\text{t}_1}\, D_1,$$

which is equivalent to

$$D_1\, L^{(1)}(v\xi)^{\text{t}_1}\, D_1^{-1}\, L^{(1)}(v) = L^{(2)}(v)\, D_2\, L^{(2)}(v\xi)^{\text{t}_2}\, D_2^{-1}.$$

This proves (3.32).

To prove that the elements $z^{\pm}(u)$ are central elements in the algebra $U_q(\tilde{\mathfrak{g}})$ we consider the product $z(u)\mathbb{I}_1\, L^{(2)}(v)$ and a chain of equalities

$$\begin{aligned}
z(u)\mathbb{I}_1\, L^{(2)}(v) &= D_1\, L^{(1)}(u\xi)^{\text{t}_1}\, D_1^{-1}\, L^{(1)}(u)L^{(2)}(v) \\
&= D_1\, L^{(1)}(u\xi)^{\text{t}_1}\, D_1^{-1}\, R_{12}(u,v)^{-1}\, L^{(2)}(v)\, L^{(1)}(u)\, R_{12}(u,v) \\
&= x(u,v\xi)\, D_1\, L^{(1)}(\xi u)^{\text{t}_1}\, R_{12}(u\xi,v)^{\text{t}_1}\, L^{(2)}(v)\, D_1^{-1}\, L^{(1)}(u)\, R_{12}(u,v) \\
&= x(u,v\xi)\, D_1\, L^{(2)}(v)\, R_{12}(u\xi,v)^{\text{t}_1}\, L^{(1)}(u\xi)^{\text{t}_1}\, D_1^{-1}\, L^{(1)}(u)\, R_{12}(u,v) \\
&= L^{(2)}(v)\, R_{12}(u,v)^{-1}\, D_1\, L^{(1)}(\xi u)^{\text{t}_1}\, D_1^{-1}\, L^{(1)}(u)\, R_{12}(u,v) \\
&= L^{(2)}(v)\, R_{12}(u,v)^{-1}\, z(u)\mathbb{I}_1\, R_{12}(u,v) = L^{(2)}(v)\, z(u)\mathbb{I}_1.
\end{aligned}$$

Equality (3.33) can be proved by expressing $\left(L^{\pm}(v)^{-1}\right)^{\text{t}}$ from (3.32) and substituting it into (3.30). $\qquad\square$

For the algebras $U_q(\tilde{\mathfrak{g}})$ with $\tilde{\mathfrak{g}} = B_n^{(1)}, C_n^{(1)}, D_n^{(1)}, A_{N-1}^{(2)}$ we set central elements $z^\pm(v) = 1$. Then equalities (3.32) take the form

$$\mathrm{D}\, \mathrm{L}^\pm(v\xi)^\mathrm{t}\, \mathrm{D}^{-1} = \mathrm{L}^\pm(v)^{-1},$$

or

$$\mathrm{D}\, \hat{\mathrm{L}}^\pm(v\xi)\, \mathrm{D}^{-1} = \mathrm{L}^\pm(v), \tag{3.34}$$

where transposed-inversed L-operators $\hat{\mathrm{L}}^\pm(u)$ are defined as

$$\hat{\mathrm{L}}^\pm(u) = \left(\mathrm{L}^\pm(u)^\mathrm{t}\right)^{-1}. \tag{3.35}$$

Due to (3.33) the central elements $Z^\pm(v)$ also equal to 1 when $z^\pm(v) = 1$. Then equality (3.30) can be written in the form

$$\hat{\mathrm{L}}^\pm(v) = \left(\mathrm{L}^\pm(u)^\mathrm{t}\right)^{-1} = \mathrm{D}^{-2}\left(\mathrm{L}^\pm(v\xi^{-2})^{-1}\right)^\mathrm{t} \mathrm{D}^2 \tag{3.36}$$

and describes the relations between order of transposition and taking inverse of the fundamental L-operators in the algebra $U_q(\tilde{\mathfrak{g}})$.

One can check that L-operators $\hat{\mathrm{L}}^\pm(u)$ given by (3.35) satisfy the same commutation relations (3.27). Let us apply to (3.27) the transposition (2.10) in both auxiliary spaces and use (2.12) to get

$$\mathrm{L}^{(1)}(u)^{\mathrm{t}_1}\, \mathrm{L}^{(2)}(v)^{\mathrm{t}_2}\, \mathrm{R}_{12}(u,v) = \mathrm{R}_{12}(u,v)\, \mathrm{L}^{(2)}(v)^{\mathrm{t}_2}\, \mathrm{L}^{(1)}(u)^{\mathrm{t}_1}.$$

Multiplying from both sides of this equality first by $\hat{\mathrm{L}}^{(1)}(u)$ and then by $\hat{\mathrm{L}}^{(2)}(v)$ one gets

$$\mathrm{R}_{12}(u,v)\, \hat{\mathrm{L}}^{(1)}(u)\, \hat{\mathrm{L}}^{(2)}(v) = \hat{\mathrm{L}}^{(2)}(v)\, \hat{\mathrm{L}}^{(1)}(u)\, \mathrm{R}_{12}(u,v).$$

Summarizing we conclude that the map

$$\mathrm{L}^\pm(u) \to \hat{\mathrm{L}}^\pm(u)$$

moves the algebra $U_q(\tilde{\mathfrak{g}})$ into the algebra given by the same commutation relation (3.27) but for the transposed-inversed L-operators $\hat{\mathrm{L}}^\pm(u)$. One can also check that central elements $\hat{Z}^\pm(v)$ and $\hat{z}^\pm(v)$ defined by (3.30) and (3.32) with $\mathrm{L}^\pm(v)$ replaced by $\hat{\mathrm{L}}^\pm(v)$ are related to the central elements $Z^\pm(v)$ and $z^\pm(v)$ as follows

$$\hat{Z}^\pm(v) = Z^\pm(v)^{-1}, \qquad \hat{z}^\pm(v) = z^\pm(v)^{-1}.$$

## 4   Gauss coordinates

It is known [16] that Gauss coordinates of L-operators introduced below by the equality (4.37) are related to the Cartan-Weyl generators of the algebra $U_q(\tilde{\mathfrak{g}})$. The Cartan-Weyl generators satisfy certain ordering properties described in details in [3] and shortly presented in the section 4.1. In this paper we consider Gauss decomposition of the fundamental L-operators of the algebra $U_q(\tilde{\mathfrak{g}})$

$$\mathrm{L}_{i,j}^\pm(u) = \sum_{\ell \leq \min(i,j)} \mathrm{F}_{j,\ell}^\pm(u)\, k_\ell^\pm(u)\, \mathrm{E}_{\ell,i}^\pm(u), \tag{4.37}$$

where one assumes that $\mathrm{F}_{i,i}^\pm(u) = \mathrm{E}_{i,i}^\pm(u) = 1$ for $1 \leq i \leq N$.

Gauss decompositions formula for the matrix entries of L-operators is associated with the products of lower triangular, diagonal and upper triangular matrices

$$\mathrm{L}^\pm(u)^\mathrm{t} = \sum_{1 \leq i,j \leq N} \mathrm{L}_{i,j}^\pm(u)\, \mathrm{e}_{ij}^\mathrm{t} = \left(\sum_{1 \leq i < j \leq N} \mathrm{e}_{ij}^\mathrm{t}\, \mathrm{F}_{j,i}^\pm(u)\right) \cdot \left(\sum_{1 \leq i \leq N} \mathrm{e}_{ii}^\mathrm{t}\, k_i^\pm(u)\right) \cdot \left(\sum_{1 \leq i < j \leq N} \mathrm{e}_{ji}^\mathrm{t}\, \mathrm{E}_{i,j}^\pm(u)\right). \tag{4.38}$$

Equality (4.38) allows to obtain Gauss decomposition of L-operators $\hat{L}^{\pm}(u)$. Indeed, using multiplication rule for $e^t_{ij} = \varepsilon_i \varepsilon_j \, e_{j'i'}$

$$e^t_{ij} \cdot e^t_{kl} = \delta_{il} \, e^t_{kj}$$

and taking inverse of both sides of the equality (4.38)

$$
\begin{aligned}
\hat{L}^{\pm}(u) = \left( \left( L^{\pm}(u) \right)^t \right)^{-1} &= \sum_{1 \leq i,j \leq N} e_{ij} \, \hat{L}^{\pm}_{i,j}(u) \\
&= \left( \sum_{1 \leq i \leq j \leq N} e^t_{ji} \, \tilde{E}^{\pm}_{i,j}(u) \right) \cdot \left( \sum_{1 \leq i \leq N} e^t_{ii} \, k^{\pm}_i(u)^{-1} \right) \cdot \left( \sum_{1 \leq i \leq j \leq N} e^t_{ij} \, \tilde{F}^{\pm}_{j,i}(u) \right)
\end{aligned}
\tag{4.39}
$$

one obtains

$$\hat{L}^{\pm}_{i,j}(u) = \varepsilon_i \varepsilon_j \sum_{\ell \leq \min(i,j)} \tilde{E}^{\pm}_{i',\ell'}(u) \, k^{\pm}_{\ell'}(u)^{-1} \, \tilde{F}^{\pm}_{\ell',j'}(u), \tag{4.40}$$

where equality $\varepsilon_i \varepsilon_j = \varepsilon_{i'} \varepsilon_{j'}$ was used. Gauss coordinates $\tilde{F}^{\pm}_{j,i}(u)$ and $\tilde{E}^{\pm}_{i,j}(u)$ in (4.39) and (4.40) satisfy recurrence relations

$$\sum_{i \leq \ell \leq j} F^{\pm}_{j,\ell}(u)\tilde{F}^{\pm}_{\ell,i}(u) = \delta_{ij} \quad \text{and} \quad \sum_{i \leq \ell \leq j} E^{\pm}_{i,\ell}(u)\tilde{E}^{\pm}_{\ell,j}(u) = \delta_{ij},$$

which can be resolved in the form

$$\tilde{F}^{\pm}_{j,i}(u) = -F^{\pm}_{j,i}(u) + \sum_{\ell=1}^{j-i-1} (-)^{\ell+1} \sum_{j > i_\ell > \cdots > i_1 > i} F^{\pm}_{j,i_\ell}(u)F^{\pm}_{i_\ell,i_{\ell-1}}(u)\cdots F^{\pm}_{i_2,i_1}(u)F^{\pm}_{i_1,i}(u)$$

and

$$\tilde{E}^{\pm}_{i,j}(u) = -E^{\pm}_{i,j}(u) + \sum_{\ell=1}^{j-i-1} (-)^{\ell+1} \sum_{j > i_\ell > \cdots > i_1 > i} E^{\pm}_{i,i_1}(u)E^{\pm}_{i_1,i_2}(u)\cdots E^{\pm}_{i_{\ell-1},i_\ell}(u)E^{\pm}_{i_\ell,j}(u).$$

## 4.1 Normal ordering of the Gauss coordinates

In order to obtain commutation relations for Gauss coordinates from (3.28) one can use the normal ordering of the Cartan-Weyl generators [3].

Let $U^{\pm}_f$, $U^{\pm}_e$ and $U^{\pm}_k$ be subalgebras of $U_q(\tilde{\mathfrak{g}})$ formed by the modes of the Gauss coordinates $F^{\pm}_{j,i}(u)$, $E^{\pm}_{i,j}(u)$ and $k^{\pm}_j(u)$, respectively. The fact that these unions of generators are subalgebras follows from the identification of modes of Gauss coordinates with Cartan-Weyl generators [16]. It is known that Cartan-Weyl generators have two natural circular orderings which imply the normal ordering of the subalgebras formed by the Gauss coordinates. These orderings are

$$\cdots \prec U^-_k \prec U^-_f \prec U^+_f \prec U^+_k \prec U^+_e \prec U^-_e \prec U^-_k \prec \cdots \tag{4.41}$$

or

$$\cdots \prec U^+_k \prec U^+_f \prec U^-_f \prec U^-_k \prec U^-_e \prec U^+_e \prec U^+_k \prec \cdots. \tag{4.42}$$

If one places subalgebras $U^{\pm}_f$, $U^{\pm}_e$ and $U^{\pm}_k$ onto circles

$$
\begin{array}{ccccccc}
& U^-_e & & U^+_e & & & U^-_e & & U^+_e \\
U^-_k & & \circlearrowleft & & U^+_k & \text{and} & U^-_k & & \circlearrowright & & U^+_k \\
& U^-_f & & U^+_f & & & U^-_f & & U^+_f
\end{array}
\tag{4.43}
$$

then ordering (4.41) is counterclockwise in the left circle and the ordering (4.42) is clockwise in the right circle of (4.43). The general theory of the Cartan-Weyl basis allows to prove that in both types of ordering the unions of subalgebras $U_f^\pm$, $U_e^\pm$ and $U_k^\pm$ along smallest arcs between starting and ending points are subalgebras in $U_q(\tilde{\mathfrak{g}})$. For example, the union of subalgebras $U_f^+ \cup U_k^+$ or $U_f^- \cup U_f^+ \cup U_k^+$ or $U_q^+(\tilde{\mathfrak{g}}) = U_f^+ \cup U_k^+ \cup U_e^+$ and so on are subalgebras in $U_q(\tilde{\mathfrak{g}})$.

The notion of the normal ordering yields a powerful practical tool to get relations for the Gauss coordinates of the specific type. In any relation which contains Gauss coordinates of the different types one first has to order all monomials according to (4.41) or (4.42) and then single out all the terms which belong to the one of subalgebras which is composed from the Gauss coordinates of the necessary type. We call this procedure *a restriction* to subalgebras in $U_q(\tilde{\mathfrak{g}})$ and will use this method to get relations between Gauss coordinates from RLL-commutation relations (3.28).

Subalgebras $U_q^\pm(\tilde{\mathfrak{g}}) = U_f^\pm \cup U_k^\pm \cup U_e^\pm$ were already introduced above as Borel subalgebras in $U_q(\tilde{\mathfrak{g}})$. To describe so called 'new realization' of these algebras in terms of the currents [5] one has to consider different types of Borel subalgebras $U_f = U_f^- \cup U_f^+ \cup U_k^+$ and $U_e = U_e^+ \cup U_e^- \cup U_k^-$. In [18] certain projections $P_f^\pm$ and $P_e^\pm$ onto intersections of the Borel subalgebras of the different types were introduced. These projections were further investigated in [3] for the ordering (4.41) and was used for the first time in [19] to describe the off-shell Bethe vectors or weight functions in terms of the Cartan-Weyl generators. One can check that the action of the projections $P_f^\pm$ and $P_e^\pm$ onto Borel subalgebras $U_f$ and $U_e$ introduced in [18] coincides with restrictions onto subalgebras $U_f^\pm$ and $U_e^\pm$ defined for the ordering (4.41).

# 5 Embedding theorem

Each algebra $\tilde{\mathfrak{g}}$ of the type $B_n^{(1)}$, $C_n^{(1)}$, $D_n^{(1)}$ and $A_{N-1}^{(2)}$ has rank $n$ as rank of the underlying finite dimensional algebra. To stress this fact we will use notation $U_q^n(\tilde{\mathfrak{g}})$ to denote explicitly rank for any of the quantum loop algebras considered in this paper. Following ideas of the paper [7] we consider in this section embedding of smaller algebras $U_q^{n-1}(\tilde{\mathfrak{g}}) \hookrightarrow U_q^n(\tilde{\mathfrak{g}})$. To note that R-matrix corresponds to the algebra $U_q^n(\tilde{\mathfrak{g}})$ we will use superscript $R^n(u,v)$, $\mathbb{R}^n(u,v)$, $\mathbb{Q}^n(u,v)$, etc.

In this paper we use Gauss decomposition of the L-operators for the algebra $U_q^n(\tilde{\mathfrak{g}})$ given by (4.37)

$$
\begin{aligned}
L_{i,j}^\pm(u) &= \sum_{1 \le \ell \le \min(i,j)} F_{j,\ell}^\pm(u)\, k_\ell^\pm(u)\, E_{\ell,i}^\pm(u) \\
&= M_{i,j}^\pm(u) + F_{j,1}^\pm(u) k_1^\pm(u) E_{1,i}^\pm(u) = M_{i,j}^\pm(u) + L_{1,j}^\pm(u) L_{1,1}^\pm(u)^{-1} L_{i,1}^\pm(u).
\end{aligned}
\tag{5.44}
$$

Let us consider matrix entries $M_{i,j}^\pm(u)$ defined by (5.44) for $1 < i,j < N$. These are matrix entries of the $(N-2) \times (N-2)$ matrix $M^\pm(u)$ of the fundamental L-operators for the algebra $U_q^{n-1}(\tilde{\mathfrak{g}})$. For $1 < i,j < N$ matrix entries $M_{i,j}^\pm(u)$ have Gauss decomposition

$$
M_{i,j}^\pm(u) = \sum_{2 \le \ell \le \min(i,j)} F_{j,\ell}^\pm(u)\, k_\ell^\pm(u)\, E_{\ell,i}^\pm(u).
\tag{5.45}
$$

For $\tilde{\mathfrak{g}} = B_n^{(1)}, C_n^{(1)}, D_n^{(1)}$ and $A_{N-1}^{(2)}$ we have following

**Theorem 5.1.** *The commutation relations for the $U_q^{n-1}(\tilde{\mathfrak{g}})$ matrix entries $M_{i,j}^\pm(u)$ follow from the Yang-Baxter equation (2.14) and the commutation relations (3.27) in $U_q^n(\tilde{\mathfrak{g}})$ and take the form* ($\mu, \rho = \pm$)

$$
R_{12}^{n-1}(u,v)\, (M^\mu(u) \otimes \mathbb{I})\, (\mathbb{I} \otimes M^\rho(v)) = (\mathbb{I} \otimes M^\rho(v))\, (M^\mu(u) \otimes \mathbb{I})\, R_{12}^{n-1}(u,v).
\tag{5.46}
$$

To prove this theorem we formulate auxiliary Lemmas 5.2 and 5.3. Let $\mathbb{L}^{(1,2)}(u)$ be fused L-operator defined as (we again skip superscripts of L-operators to avoid bulky notations)

$$\mathbb{L}^{(1,2)}(u) = R(1, q^2)\, L^{(1)}(u)\, L^{(2)}(q^2 u) = L^{(2)}(q^2 u)\, L^{(1)}(u)\, R(1, q^2)\,.$$

One can calculate its $(i, j; 1, 1)$ matrix element

$$
\begin{aligned}
\mathbb{L}_{i,j;1,1}(u) &= \langle i, 1|\mathbb{L}^{(1,2)}(u)|j, 1\rangle = \sum_{k,l=1}^{N} R_{i,k;1,l}(1, q^2) L_{k,j}(u) L_{l,1}(q^2 u) \\
&= L_{i,j}(u) L_{1,1}(q^2 u) - q L_{1,j}(u) L_{i,1}(q^2 u)\,,
\end{aligned}
\tag{5.47}
$$

where $|i, j\rangle = |i\rangle \otimes |j\rangle$ and $\langle i, j| = \langle i| \otimes \langle j|$ are vectors in $\left(\mathbb{C}^N\right)^{\otimes 2}$ such that $\langle i|j\rangle = \delta_{ij}$.

The commutation relations for $1 < i < N$

$$L_{1,1}(u)^{-1} L_{i,1}(u) = q\, L_{i,1}(q^2 u) L_{1,1}(q^2 u)^{-1}$$

and (5.47) imply that L-operators $M(u)$ for the algebra $U_q^{n-1}(\tilde{\mathfrak{g}})$ can be presented as

$$M_{i,j}(u) = \mathbb{L}_{i,j;1,1}(u)\, L_{1,1}(q^2 u)^{-1}\,,
\tag{5.48}$$

where $1 < i, j < N$.

**Lemma 5.2.** *There is a commutativity of the matrix entries in* $U_q^n(\tilde{\mathfrak{g}})$

$$L_{1,1}(u)\, M_{i,j}(v) = M_{i,j}(v)\, L_{1,1}(u)\,, \quad 1 < i, j < N\,.$$

According to (5.48) matrix entries $M_{i,j}(u)$ are proportional to the matrix entries $\mathbb{L}_{i,j;1,1}(u)$ up to commuting with $\mathbb{L}_{i,j;1,1}(u)$ invertible operator $L_{1,1}(v)$ (see Appendix A). It yields that the commutation relations for $M_{i,j}(u)$ should coincide with the commutation relations of $\mathbb{L}_{i,j;1,1}(u)$. To find the commutation relations for the matrix entries $\mathbb{L}_{i,j;1,1}(u)$ we need following

**Lemma 5.3.** *There are equalities for* $1 < i, j < N$

$$R_{12}^n(1, q^2) R_{34}^n(1, q^2) R_{14}^n(u, q^2 v) R_{13}^n(u, v)|i, 1, j, 1\rangle = R_{12}^n(1, q^2) R_{34}^n(1, q^2) R_{13}^{n-1}(u, v)|i, 1, j, 1\rangle \tag{5.49}$$

*and*

$$\langle i, 1, j, 1|R_{13}^n(u, v) R_{14}^n(u, q^2 v) R_{34}^n(1, q^2) R_{12}^n(1, q^2) = \langle i, 1, j, 1|R_{13}^{n-1}(u, v) R_{34}^n(1, q^2) R_{12}^n(1, q^2)\,, \tag{5.50}$$

*where* $|i, k, j, l\rangle = |i\rangle \otimes |k\rangle \otimes |j\rangle \otimes |l\rangle$ *and* $\langle i, k, j, l| = \langle i| \otimes \langle k| \otimes \langle j| \otimes \langle l|$ *are vectors in* $\left(\mathbb{C}^N\right)^{\otimes 4}$.

Proofs of the Lemmas 5.2 and 5.3 can be found in Appendix A.

To prove theorem 5.1 we consider RLL-commutation relations for L-operators $\mathbb{L}(u)$ and $\mathbb{L}(v)$ (A.80) and for $1 < i_1, j_1, i_2, j_2 < N$ take the matrix element of this commutation relation

$$
\begin{aligned}
&\langle i_1, 1, j_1, 1|R_{23}^n(q^2 u, v) R_{13}^n(u, v) R_{24}^n(u, v) R_{14}^n(u, q^2 v) R_{12}^n(1, q^2) R_{34}^n(1, q^2) \\
&\quad \times L^{(1)}(u) L^{(2)}(q^2 u) L^{(3)}(v) L^{(4)}(q^2 v)|i_2, 1, j_2, 1\rangle \\
&= \langle i_1, 1, j_1, 1|L^{(3)}(v) L^{(4)}(q^2 v) L^{(1)}(u) L^{(2)}(q^2 u) \\
&\quad \times R_{34}^n(1, q^2) R_{12}^n(1, q^2) R_{14}^n(u, q^2 v) R_{24}^n(u, v) R_{13}^n(u, v) R_{23}^n(q^2 u, v)|i_2, 1, j_2, 1\rangle\,.
\end{aligned}
\tag{5.51}
$$

Let us transform last line in (5.51) using Lemma 5.3, equality (A.81) and Yang-Baxter equation (2.14). We have

$$
\begin{aligned}
& R^n_{34}(1,q^2)R^n_{12}(1,q^2)R^n_{14}(u,q^2v)R^n_{24}(u,v)R^n_{13}(u,v)R^n_{23}(q^2u,v)|i_2,1,j_2,1\rangle \\
&= R^n_{12}(1,q^2)R^n_{13}(u,v)R^n_{14}(u,q^2v)R^n_{34}(1,q^2)R^n_{24}(u,v)R^n_{23}(q^2u,v)|i_2,1,j_2,1\rangle \\
&= f(q^2u,v)R^n_{12}(1,q^2)R^n_{34}(1,q^2)R^n_{14}(u,q^2v)R^n_{13}(u,v)|i_2,1,j_2,1\rangle \\
&= f(q^2u,v)R^n_{12}(1,q^2)R^n_{34}(1,q^2)R^{n-1}_{13}(u,v)|i_2,1,j_2,1\rangle .
\end{aligned}
\tag{5.52}
$$

At the second step of this calculation we used equality (A.81) taken at $u \to q^2u$ and scaling invariance of R-matrix (2.11).

Analogously first line in (5.51) can be transformed to

$$
\begin{aligned}
& \langle i_1,1,j_1,1|R^n_{23}(q^2u,v)R^n_{13}(u,v)R^n_{24}(u,v)R^n_{14}(u,q^2v)R^n_{12}(1,q^2)R^n_{34}(1,q^2) \\
&= f(q^2u,v)\langle i_1,1,j_1,1|R^{n-1}_{13}(u,v)R^n_{12}(1,q^2)R^n_{34}(1,q^2),
\end{aligned}
\tag{5.53}
$$

where we used (A.82) at $v \to q^{-2}v$.

Equalities (5.52) and (5.53) allow to rewrite (5.51) in the form

$$
\begin{aligned}
& \langle i_1,1,j_1,1|R^{n-1}_{13}(u,v)\, \mathbb{L}^{(1,2)}(u)\mathbb{L}^{(3,4)}(v)|i_2,1,j_2,1\rangle \\
&= \langle i_1,1,j_1,1|\mathbb{L}^{(3,4)}(v)\mathbb{L}^{(1,2)}(u)\, R^{n-1}_{13}(u,v)|i_2,1,j_2,1\rangle ,
\end{aligned}
$$

which proves the statement of theorem (5.46) due to Lemma 5.2 and relation (5.48). $\qquad\square$

Theorem 5.1 implies that in order to find the commutation relations between Gauss coordinates in the algebra $U^n_q(\tilde{\mathfrak{g}})$ it is sufficient to obtain these commutation relations for the smallest rank nontrivial algebras. We will find such commutation relations in the algebras $U_q(\tilde{\mathfrak{g}})$ for $\tilde{\mathfrak{g}}$ of the types $B^{(1)}_n$, $C^{(1)}_n$, $D^{(1)}_n$, $A^{(2)}_{N-1}$ in Appendix C.

We can formulate analogous statement for the algebra $U_q(A^{(1)}_{N-1})$. Let $\mathrm{M}_{i,j}(u)$ for $1 < i, j \le N$ be matrix entries of the L-operators for the algebra $U_q(A^{(1)}_{N-2})$ defined by (5.45). Denote by $\mathbb{R}^N(u,v)$ R-matrix (2.8) for the algebra $U_q(A^{(1)}_{N-1})$. Using similar arguments as above we can prove following

**Proposition 5.4.** *The commutation relations of the matrix entries $\mathrm{M}_{i,j}(u)$ and their Gauss coordinates of the fundamental L-operators of the algebra $U_q(A^{(1)}_{N-2})$ for $1 < i, j \le N$ follow from the commutation relations (3.27) for the algebra $U_q(A^{(1)}_{N-1})$ and take the same form with R-matrix $\mathbb{R}^{N-1}(u,v)$.*

We are not going to provide a proof of this Proposition since it can be performed in a similar way as the proof of theorem 5.1. Practical meaning of this Proposition is that in order to obtain the commutation relations between Gauss coordinates for the algebra $U_q(A^{(1)}_{N-1})$ it is sufficient to consider the commutation relations for the algebras at small values of $N$.

## 5.1 Embedding in terms of the Gauss coordinates

Let us introduce 'alternative' to (4.37) Gauss decomposition of the fundamental L-operators

$$
\mathrm{L}(u) = \left(\sum_{1 \le i \le j \le N} e_{ji}\, \bar{\mathrm{E}}_{i,j}(q^{-2(i-1)}u)\right) \times \left(\sum_{1 \le i \le N} e_{ii}\, \bar{k}_i(q^{-2(i-1)}u)\right) \cdot \left(\sum_{1 \le i \le j \le N} e_{ij}\, \bar{\mathrm{F}}_{j,i}(q^{-2(i-1)}u)\right),
$$

where shifts by $q^{-2(i-1)}$ in the arguments of 'alternative' Gauss coordinates $\bar{F}_{j,i}(u)$, $\bar{E}_{i,j}(u)$ and $\bar{k}_i(u)$ are introduced for the further convenience. In terms of these Gauss coordinates matrix entries of L-operators have the form

$$L_{i,j}(u) = \sum_{1 \le \ell \le \min(i,j)} \bar{E}_{\ell,i}(q^{-2(\ell-1)}u)\, \bar{k}_\ell(q^{-2(\ell-1)}u)\, \bar{F}_{j,\ell}(q^{-2(\ell-1)}u). \tag{5.54}$$

In (5.54) we assume that $\bar{F}_{i,i}^\pm(u) = \bar{E}_{i,i}^\pm(u) = 1$ for $1 \le i \le N$.

Our goal is to find relations between Gauss coordinates $F_{j,i}(u)$, $E_{i,j}(u)$, $k_j(u)$ and $\bar{F}_{j,i}(u)$, $\bar{E}_{i,j}(u)$, $\bar{k}_j(u)$. This is given by

**Proposition 5.5.** *For* $\tilde{\mathfrak{g}} = B_n^{(1)}, C_n^{(1)}, D_n^{(1)}, A_{N-1}^{(2)}$ *Gauss coordinates* $\bar{F}_{j,i}^\pm(u)$, $\bar{E}_{i,j}^\pm(u)$ *and* $\bar{k}_j^\pm(u)$ *are related to the initial Gauss coordinates* $F_{j,i}^\pm(u)$, $E_{i,j}^\pm(u)$ *and* $k_j^\pm(u)$:

$$\bar{F}_{j,i}^\pm(u) = q F_{j,i}^\pm(q^{-2}u), \tag{5.55}$$

$$\bar{E}_{i,j}^\pm(u) = q^{-1} E_{i,j}^\pm(q^{-2}u), \tag{5.56}$$

$$\bar{k}_\ell^\pm(u) = k_\ell^\pm(u) \prod_{s=1}^{\ell-1} \frac{k_s^\pm(q^{2(\ell-s)}u)}{k_s^\pm(q^{2(\ell-s-1)}u)}, \tag{5.57}$$

*where* $i < j < i'$, $1 \le i \le n$, $1 \le \ell \le n+1$ *for odd* $N = 2n+1$ *and* $1 \le i \le n-1$, $1 \le \ell \le n$ *for* $N = 2n$ *even. Moreover, for* $1 \le i < j \le N$ *and* $1 \le \ell \le N$

$$\bar{F}_{j,i}^\pm(u) = \varepsilon_i \varepsilon_j\, q^{\bar{i}-\bar{j}} \check{F}_{i',j'}^\pm(q^{2(i-1)}\xi u), \tag{5.58}$$

$$\bar{E}_{i,j}^\pm(u) = \varepsilon_i \varepsilon_j\, q^{\bar{j}-\bar{i}} \check{E}_{j',i'}^\pm(q^{2(i-1)}\xi u), \tag{5.59}$$

$$\bar{k}_\ell^\pm(u) = k_{\ell'}^\pm(q^{2(\ell-1)}\xi u)^{-1}. \tag{5.60}$$

The proof of this Proposition is given in Appendix B.

Note that equality (5.57) can be written in the form

$$\bar{k}_\ell^\pm(u) = k_\ell^\pm(u) \frac{\bar{k}_{\ell-1}^\pm(q^2 u)}{k_{\ell-1}^\pm(u)},$$

which is a consequence of the embedding relation (B.91) at each step of the embedding. Moreover, we can exclude $\bar{k}_\ell^\pm(u)$ from (5.57) and (5.60) to obtain

$$k_{\ell'}^\pm(u) = k_\ell^\pm(q^{-2(\ell-1)}\xi^{-1}u)^{-1} \prod_{s=1}^{\ell-1} \frac{k_s^\pm(q^{-2s}\xi^{-1}u)}{k_s^\pm(q^{2(1-s)}\xi^{-1}u)}, \tag{5.61}$$

where $1 \le \ell \le n+1$ for odd $N = 2n+1$ and $1 \le \ell \le n$ for $N = 2n$ even.

Proposition 5.5 has an obvious

**Corollary 5.6.** *There are relations between Gauss coordinates of the fundamental* L-*operators in the algebra* $U_q(\tilde{\mathfrak{g}})$ *corresponding to the simple roots of the underlying algebra* $\mathfrak{g}$

$$\begin{aligned}
F_{N+1-i,N-i}^\pm(u) &= -F_{i+1,i}^\pm(q^{-2i}\xi^{-1}u), \\
E_{N-i,N+1-i}^\pm(u) &= -E_{i,i+1}^\pm(q^{-2i}\xi^{-1}u),
\end{aligned} \tag{5.62}$$

*for* $\forall N$ *and* $1 \le i \le n-1$ *and*

$$\begin{aligned}
F_{n+2,n+1}^\pm(u) &= -q^{1/2}\, F_{n+1,n}^\pm(q^{-2n}\xi^{-1}u), \\
E_{n+1,n+2}^\pm(u) &= -q^{-1/2}\, E_{n,n+1}^\pm(q^{-2n}\xi^{-1}u),
\end{aligned} \tag{5.63}$$

*for* $N = 2n+1$.

This corollary together with equalities (5.61) defines the algebraically independent sets of the generators in each of the algebras $U_q(\tilde{\mathfrak{g}})$ of the type $\tilde{\mathfrak{g}} = B_n^{(1)}, C_n^{(1)}, D_n^{(1)}$ and $A_{N-1}^{(2)}$.

# 6 New realization of the algebra $U_q(\tilde{\mathfrak{g}})$

A new realization of the quantum affine algebras $U_q(\hat{\mathfrak{g}})$ was given in [5] in terms of the formal series called *currents* labeled by the simple roots of the underlying finite-dimensional algebra $\mathfrak{g}$. Relations between currents and Gauss coordinates for the algebra $\tilde{U}_q(A_{N-1}^{(1)})$ was given in [6]

$$
\begin{aligned}
F_i(u) &= \mathrm{F}_{i+1,i}^+(u) - \mathrm{F}_{i+1,i}^-(u) = \sum_{\ell \in \mathbb{Z}} \mathrm{sign}(\ell) \mathrm{F}_{i+1,i}[\ell] u^{-\ell}, \\
E_i(u) &= \mathrm{E}_{i,i+1}^+(u) - \mathrm{E}_{i,i+1}^-(u) = -\sum_{\ell \in \mathbb{Z}} \mathrm{sign}(-\ell) \mathrm{E}_{i,i+1}[\ell] u^{-\ell},
\end{aligned}
\tag{6.64}
$$

where $1 \le i \le N-1$.

For the algebras $U_q(B_n^{(1)})$, $U_q(C_n^{(1)})$ and $U_q(A_{N-1}^{(2)})$ the currents are introduced by the formulas (6.64) for $1 \le i \le n$. For the algebra $U_q(D_n^{(1)})$ first $(n-1)$ currents are also introduced by (6.64) with $1 \le i \le n-1$ and the currents $F_n(u)$ and $E_n(u)$ by the equalities [8]

$$
\begin{aligned}
F_n(u) &= \mathrm{F}_{n+1,n-1}^+(u) - \mathrm{F}_{n+1,n-1}^-(u) = \mathrm{F}_{n+2,n}^-(u) - \mathrm{F}_{n+2,n}^+(u), \\
E_n(u) &= \mathrm{E}_{n-1,n+1}^+(u) - \mathrm{E}_{n-1,n+1}^-(u) = \mathrm{E}_{n,n+2}^-(u) - \mathrm{E}_{n,n+2}^+(u).
\end{aligned}
\tag{6.65}
$$

It is obvious from the commutation relations in the algebra $U_q(\tilde{\mathfrak{g}})$ (3.28) that term $\mathbb{Q}(u,v)$ of the R-matrix (2.9) do not contribute into commutation relations of the matrix entries $\mathrm{L}_{i,j}^\pm(u)$ and $\mathrm{L}_{k,l}^\pm(v)$ for $1 \le i,j,k,l \le n$. Commutation relations between these matrix entries and between corresponding Gauss coordinates are defined by $U_q(\mathfrak{gl}_n)$-invariant R-matrix (2.8). These commutation relations can be translated into commutation relations between currents $F_i(u)$, $E_i(u)$, $1 \le i \le n-1$ and Gauss coordinates $k_\ell^\pm(u)$, $1 \le \ell \le n$ according to the standard approach developed in [6]. We formulate all nontrivial commutation relations between these currents without proofs

$$
\begin{aligned}
k_i^\pm(u) F_i(v) k_i^\pm(u)^{-1} &= \frac{q^{-1}u - qv}{u-v} F_i(v), \\
k_{i+1}^\pm(u) F_i(v) k_{i+1}^\pm(u)^{-1} &= \frac{qu - q^{-1}v}{u-v} F_i(v), \\
k_i^\pm(u)^{-1} E_i(v) k_i^\pm(u) &= \frac{q^{-1}u - qv}{u-v} E_i(v), \\
k_{i+1}^\pm(u)^{-1} E_i(v) k_{i+1}^\pm(u) &= \frac{qu - q^{-1}v}{u-v} E_i(v), \\
(q^{-1}u - qv) F_i(u) F_i(v) &= (qu - q^{-1}v) F_i(v) F_i(u), \\
(qu - q^{-1}v) E_i(u) E_i(v) &= (q^{-1}u - qv) E_i(v) E_i(u), \\
(u-v) F_i(u) F_{i+1}(v) &= (q^{-1}u - qv) F_{i+1}(v) F_i(u), \\
(q^{-1}u - qv) E_i(u) E_{i+1}(v) &= (u-v) E_{i+1}(v) E_i(u), \\
[E_i(u), F_j(v)] &= \delta_{i,j} (q - q^{-1}) \delta(u,v) \Big( k_{i+1}^-(v) k_i^-(v)^{-1} - k_{i+1}^+(u) k_i^+(u)^{-1} \Big).
\end{aligned}
\tag{6.66}
$$

There are also Serre relations for the currents $E_i(u)$ and $F_i(u)$ [5,6]

$$
\begin{aligned}
\mathrm{Sym}_{v_1,v_2} \Big[ F_i(v_1), [F_i(v_2), F_{i\pm 1}(u)]_{q^{-1}} \Big]_q &= 0, \\
\mathrm{Sym}_{v_1,v_2} \Big[ E_i(v_1), [E_i(v_2), E_{i\pm 1}(u)]_q \Big]_{q^{-1}} &= 0,
\end{aligned}
\tag{6.67}
$$

where $[A, B]_q$ means $q$-commutator

$$[A, B]_q = AB - q\, BA$$

and $\mathrm{Sym}_{v_1, v_2}\, G(v_1, v_2) \equiv G(v_1, v_2) + G(v_2, v_1)$.

The multiplicative delta function in (6.66) is defined by the formal series

$$\delta(u, v) = \sum_{\ell \in \mathbb{Z}} \frac{u^\ell}{v^\ell},$$

which satisfy the property

$$\delta(u, v)G(u) = \delta(u, v)G(v),$$

for any formal series $G(u)$.

**Remark 6.1.** The equalities (6.66) should be understood in a sense of equalities between formal series. It means that these commutation relations should be understood as infinite set of equalities between modes of the currents which appear after equating the coefficients at all powers $u^\ell v^{\ell'}$ for $\ell, \ell' \in \mathbb{Z}$. The rational functions in the commutation relations (6.66) should be understood as series over powers of $v/u$ in the relations containing the current $k_j^+(u)$ and over powers of $u/v$ in the relations with the current $k_j^-(u)$.

The commutation relations of the currents $F_n(u)$, $E_n(u)$ and diagonal Gauss coordinates will be specific for each of the algebras $U_q(B_n^{(1)})$, $U_q(C_n^{(1)})$, $U_q(D_n^{(1)})$ and $U_q(A_{N-1}^{(2)})$. According to the theorem 5.1 these commutation relations can be obtained by considering algebras of the small rank presented in the Appendix C.

## 6.1 New realization of the algebra $U_q(B_n^{(1)})$

The full set of the nontrivial commutation relation for the algebra $U_q(B_n^{(1)})$ is given by the relations (6.66) and [8]

$$k_n^\pm(u)F_n(v)k_n^\pm(u)^{-1} = \frac{q^{-1}u - qv}{u - v}\, F_n(v),$$

$$k_{n+1}^\pm(u)F_n(v)k_{n+1}^\pm(u)^{-1} = \frac{q^{-1}u - qv}{u - v}\, \frac{qu - v}{u - qv}\, F_n(v),$$

$$k_n^\pm(u)^{-1}E_n(v)k_n^\pm(u) = \frac{q^{-1}u - qv}{u - v}\, E_n(v),$$

$$k_{n+1}^\pm(u)^{-1}E_n(v)k_{n+1}^\pm(u) = \frac{q^{-1}u - qv}{u - v}\, \frac{qu - v}{u - qv}\, E_n(v),$$

$$(u - qv)\, F_n(u)F_n(v) = (qu - v)\, F_n(v)F_n(u),$$

$$(qu - v)\, E_n(u)E_n(v) = (u - qv)\, E_n(v)E_n(u),$$

$$(u - v)\, F_{n-1}(u)F_n(v) = (q^{-1}u - qv)\, F_n(v)F_{n-1}(u),$$

$$(q^{-1}u - qv)\, E_{n-1}(u)E_n(v) = (u - v)\, E_n(v)E_{n-1}(u),$$

$$[E_n(u), F_n(v)] = (q - q^{-1})\delta(u, v)\Big(k_{n+1}^-(v)\, k_n^-(v)^{-1} - k_{n+1}^+(u)\, k_n^+(u)^{-1}\Big),$$

where modes of the dependent currents $k_{n+1}^\pm(u)$ are defined by the relation

$$\prod_{\ell=1}^n k_\ell^\pm(q^{2(n-\ell)}u) = k_{n+1}^\pm(qu)\, k_{n+1}^\pm(u) \prod_{\ell=1}^n k_\ell^\pm(q^{2(n-\ell+1)}u)$$

following from (5.57) and (5.60) for $\ell = n + 1$ and $\xi = q^{1-2n}$. Serre relations which include currents $E_n(u)$ and $F_n(u)$ are

$$\operatorname*{Sym}_{v_1,v_2}\Big[F_{n-1}(v_1),[F_{n-1}(v_2),F_n(u)]_{q^{-1}}\Big]_q = 0\,,$$

$$\operatorname*{Sym}_{v_1,v_2}\Big[E_{n-1}(v_1),[E_{n-1}(v_2),E_n(u)]_q\Big]_{q^{-1}} = 0\,,$$

$$\operatorname*{Sym}_{v_1,v_2,v_3}\Big[F_n(v_1),\Big[F_n(v_2),[F_n(v_3),F_{n-1}(u)]_{q^{-1}}\Big]_q\Big] = 0\,,$$

$$\operatorname*{Sym}_{v_1,v_2,v_3}\Big[E_n(v_1),\Big[E_n(v_2),[E_n(v_3),E_{n-1}(u)]_q\Big]_{q^{-1}}\Big] = 0\,. \tag{6.68}$$

Here $\operatorname{Sym}_{v_1,v_2,v_3} G(v_1,v_2,v_2) \equiv \sum_{\sigma \in \mathfrak{S}_3} G(v_{\sigma(1)},v_{\sigma(2)},v_{\sigma(3)})$.

## 6.2 New realization of the algebra $U_q(C_n^{(1)})$

The nontrivial commutation relations for the set of the currents in the algebra $U_q(C_n^{(1)})$ are given by the commutation relations (6.66) and the commutation relations involving the currents $F_n(u)$ and $E_n(u)$ [7]

$$k_n^{\pm}(u)F_n(v)k_n^{\pm}(u)^{-1} = \frac{q^{-2}u - q^2 v}{u - v}\,F_n(v)\,,$$

$$k_n^{\pm}(u)^{-1}E_n(v)k_n^{\pm}(u) = \frac{q^{-2}u - q^2 v}{u - v}\,E_n(v)\,,$$

$$(q^{-2}u - q^2 v)\,F_n(u)F_n(v) = (q^2 u - q^{-2}v)\,F_n(v)F_n(u)\,,$$

$$(q^2 u - q^{-2}v)\,E_n(u)E_n(v) = (q^{-2}u - q^2 v)\,E_n(v)E_n(u)\,,$$

$$(u - v)\,F_{n-1}(u)F_n(v) = (q^{-2}u - q^2 v)\,F_n(v)F_{n-1}(u)\,,$$

$$(q^{-2}u - q^2 v)\,E_{n-1}(u)E_n(v) = (u - v)\,E_n(v)E_{n-1}(u)\,,$$

$$[E_n(u),F_n(v)] = (q^2 - q^{-2})\delta(u,v)\Big(k_{n+1}^-(u)\cdot k_n^-(u)^{-1} - k_{n+1}^+(v)\cdot k_n^+(v)^{-1}\Big)\,,$$

where

$$k_{n+1}^{\pm}(u) = k_n^{\pm}(q^4 u)^{-1}\prod_{\ell=1}^{n-1}\frac{k_\ell^{\pm}(q^{2n+2-2\ell}u)}{k_\ell^{\pm}(q^{2n+4-2\ell}u)}\,.$$

Serre relations which include the currents $F_n(u)$ and $E_n(u)$ are

$$\operatorname*{Sym}_{v_1,v_2}\Big[F_n(v_1),[F_n(v_2),F_{n-1}(u)]_{q^{-2}}\Big]_{q^2} = 0\,,$$

$$\operatorname*{Sym}_{v_1,v_2}\Big[E_n(v_1),[E_n(v_2),E_{n-1}(u)]_{q^2}\Big]_{q^{-2}} = 0\,,$$

$$\operatorname*{Sym}_{v_1,v_2,v_3}\Big[F_{n-1}(v_1),\Big[F_{n-1}(v_2),[F_{n-1}(v_3),F_n(u)]_{q^{-2}}\Big]_{q^2}\Big] = 0\,,$$

$$\operatorname*{Sym}_{v_1,v_2,v_3}\Big[E_{n-1}(v_1),\Big[E_{n-1}(v_2),[E_{n-1}(v_3),E_n(u)]_{q^2}\Big]_{q^{-2}}\Big] = 0\,. \tag{6.69}$$

## 6.3 New realization of the algebra $U_q(D_n^{(1)})$

Using results presented in Appendix C.3 one can obtain that the current realization of the algebra $U_q(D_n^{(1)})$ is given by the commutation relations (6.66) and all nontrivial commutation

relations which include the currents $F_n(u)$ and $E_n(u)$ are [8]

$$k^{\pm}_{n-1}(u)F_n(v)k^{\pm}_{n-1}(u)^{-1} = \frac{q^{-1}u - qv}{u - v} F_n(v),$$

$$k^{\pm}_{n}(u)F_n(v)k^{\pm}_{n}(u)^{-1} = \frac{q^{-1}u - qv}{u - v} F_n(v),$$

$$k^{\pm}_{n-1}(u)^{-1}E_n(v)k^{\pm}_{n-1}(u) = \frac{q^{-1}u - qv}{u - v} E_n(v),$$

$$k^{\pm}_{n}(u)^{-1}E_n(v)k^{\pm}_{n}(u) = \frac{q^{-1}u - qv}{u - v} E_n(v), \tag{6.70}$$

$$(q^{-1}u - qv)\, F_n(u)F_n(v) = (qu - q^{-1}v)\, F_n(v)F_n(u),$$

$$(qu - q^{-1}v)\, E_n(u)E_n(v) = (q^{-1}u - qv)\, E_n(v)E_n(u),$$

$$(u - v)\, F_{n-2}(u)F_n(v) = (q^{-1}u - qv)\, F_n(v)F_{n-2}(u),$$

$$(q^{-1}u - qv)\, E_{n-2}(u)E_n(v) = (u - v)\, E_n(v)E_{n-2}(u),$$

$$[E_n(u), F_n(v)] = (q - q^{-1})\delta(u,v)\left(k^{-}_{n+1}(u) \cdot k^{-}_{n-1}(u)^{-1} - k^{+}_{n+1}(v) \cdot k^{+}_{n-1}(v)^{-1}\right).$$

In (6.70) the Gauss coordinates $k^{\pm}_{n+1}(u)$ are given by (5.61) for $\ell = n$ and $\xi = q^{2-2n}$. The Serre relations which include currents $F_n(u)$ and $E_n(u)$ can be written in the form [8]

$$\underset{v_1,v_2}{\mathrm{Sym}}\left[F_i(v_1),[F_i(v_2),F_n(u)]_{q^{-1}}\right]_q = 0, \quad \underset{v_1,v_2}{\mathrm{Sym}}\left[F_n(v_1),[F_n(v_2),F_i(u)]_{q^{-1}}\right]_q = 0,$$

$$\underset{v_1,v_2}{\mathrm{Sym}}\left[E_i(v_1),[E_i(v_2),E_n(u)]_q\right]_{q^{-1}} = 0, \quad \underset{v_1,v_2}{\mathrm{Sym}}\left[E_n(v_1),[E_n(v_2),E_i(u)]_q\right]_{q^{-1}} = 0,$$

for $i = n-2, n-1$.

## 6.4 New realization of the algebra $U_q(A^{(2)}_{2n})$

Nontrivial commutation relations for the new realization of the algebra $U_q(A^{(2)}_{2n})$ is given by the relations (6.66) and additional relations which include currents $F_n(u)$ and $E_n(u)$ listed below

$$k^{\pm}_{n}(u)F_n(v)k^{\pm}_{n}(u)^{-1} = \frac{q^{-1}u - qv}{u - v} F_n(v),$$

$$k^{\pm}_{n+1}(u)F_n(v)k^{\pm}_{n+1}(u)^{-1} = \frac{qu - q^{-1}v}{u - v}\frac{qv + u}{v + qu} F_n(v),$$

$$k^{\pm}_{n}(u)^{-1}E_n(v)k^{\pm}_{n}(u) = \frac{q^{-1}u - qv}{u - v} E_n(v),$$

$$k^{\pm}_{n+1}(u)^{-1}E_n(v)k^{\pm}_{n+1}(u) = \frac{qu - q^{-1}v}{u - v}\frac{qv + u}{v + qu} E_n(v),$$

$$(q^{-1}u - qv)(qu + v)\, F_n(u)F_n(v) = (qu - q^{-1}v)(u + qv)\, F_n(v)F_n(u),$$

$$(qu - q^{-1}v)(u + qv)\, E_n(u)E_n(v) = (q^{-1}u - qv)(qu + v)\, E_n(v)E_n(u),$$

$$(u - v)\, F_{n-1}(u)F_n(v) = (q^{-1}u - qv)\, F_n(v)F_{n-1}(u),$$

$$(q^{-1}u - qv)\, E_{n-1}(u)E_n(v) = (u - v)\, E_n(v)E_{n-1}(u),$$

$$[E_n(u), F_n(v)] = (q - q^{-1})\delta(u,v)\left(k^{-}_{n+1}(v)\, k^{-}_{n}(v)^{-1} - k^{+}_{n+1}(u)\, k^{+}_{n}(u)^{-1}\right),$$

where $k^{\pm}_{n+1}(u)$ are defined by the relation

$$k^{\pm}_{n+1}(u)k^{\pm}_{n+1}(-qu) = \prod_{s=1}^{n}\frac{k^{\pm}_{\ell}(-q^{2n-2s+1}u)}{k^{\pm}_{\ell}(-q^{2n-2s+3}u)}$$

following from (5.61) at $\ell = n + 1$ and $\xi = -q^{-1-2n}$. Serre relations which include currents $E_n(u)$ and $F_n(u)$ are the same as in the case of $U_q(B_n^{(1)})$ (6.68)

$$\operatorname*{Sym}_{v_1, v_2}\Big[F_{n-1}(v_1), [F_{n-1}(v_2), F_n(u)]_{q^{-1}}\Big]_q = 0\,,$$

$$\operatorname*{Sym}_{v_1, v_2}\Big[E_{n-1}(v_1), [E_{n-1}(v_2), E_n(u)]_q\Big]_{q^{-1}} = 0\,,$$

$$\operatorname*{Sym}_{v_1, v_2, v_3}\Big[F_n(v_1), \Big[F_n(v_2), [F_n(v_3), F_{n-1}(u)]_{q^{-1}}\Big]_q\Big] = 0\,,$$

$$\operatorname*{Sym}_{v_1, v_2, v_3}\Big[E_n(v_1), \Big[E_n(v_2), [E_n(v_3), E_{n-1}(u)]_q\Big]_{q^{-1}}\Big] = 0\,,$$

and there are additional Serre relations for the currents $E_n(u)$ and $F_n(u)$ which can be presented in the form [5]

$$\operatorname*{Sym}_{u, v, w}\big(u - (q + q^2)v + q^3 w\big)F_n(u)F_n(v)F_n(w) = 0\,,$$

$$\operatorname*{Sym}_{u, v, w}\big(q^3 vw - (q + q^2)uw + uv\big)F_n(u)F_n(v)F_n(w) = 0\,,$$

$$\operatorname*{Sym}_{u, v, w}\big(q^3 u - (q + q^2)v + w\big)E_n(u)E_n(v)E_n(w) = 0\,,$$

$$\operatorname*{Sym}_{u, v, w}\big(vw - (q + q^2)uw + q^3 uv\big)E_n(u)E_n(v)E_n(w) = 0\,.$$

## 6.5 New realization of the algebra $U_q(A_{2n-1}^{(2)})$

Finally using results presented in Appendix C.5 one can describe the new realization of the algebra $U_q(A_{2n-1}^{(2)})$ as collection of the commutation relations (6.66) and additional relations which include currents $F_n(u)$ and $E_n(u)$

$$k_n^{\pm}(u)F_n(v)k_n^{\pm}(u)^{-1} = \frac{q^2 v^2 - q^{-2}u^2}{v^2 - u^2}F_n(v)\,,$$

$$k_n^{\pm}(u)^{-1}E_n(v)k_n^{\pm}(u) = \frac{q^2 v^2 - q^{-2}u^2}{v^2 - u^2}E_n(v)\,,$$

$$(q^{-2}u^2 - q^2 v^2)\,F_n(u)F_n(v) = (q^2 u^2 - q^{-2}v^2)\,F_n(v)F_n(u)\,,$$

$$(q^2 u^2 - q^{-2}v^2)\,E_n(u)E_n(v) = (q^{-2}u^2 - q^2 v^2)\,E_n(v)E_n(u)\,,$$

$$(u^2 - v^2)\,F_{n-1}(u)F_n(v) = (q^{-2}u^2 - q^2 v^2)\,F_n(v)F_{n-1}(u)\,,$$

$$(q^{-2}u^2 - q^2 v^2)\,E_{n-1}(u)E_n(v) = (u^2 - v^2)\,E_n(v)E_{n-1}(u)\,,$$

$$[E_n(u), F_n(v)] = (q^2 - q^{-2})\bar{\delta}(u, v)\big(k_{n+1}^-(v)k_n^-(v)^{-1} - k_{n+1}^+(u)k_n^+(u)^{-1}\big)\,,$$

where $\delta$-function $\bar{\delta}(u, v)$ is given by the series

$$\bar{\delta}(u, v) = \sum_{\ell \in \mathbb{Z}}\frac{u^{2\ell+1}}{v^{2\ell+1}}\,.$$

The currents $F_n(u)$ and $E_n(u)$ are series with respect to the odd powers of the spectral parameters and the ratio of the Gauss coordinates $k_{n+1}^{\pm}(u)k_n^{\pm}(u)^{-1}$ are series with respect to even powers of the spectral parameters. Serre relations which include the currents $F_n(u)$ and $E_n(u)$

are the same as in the case of $U_q(C_n^{(1)})$ (6.69),

$$\underset{\nu_1,\nu_2}{\text{Sym}}\Big[F_n(\nu_1),[F_n(\nu_2),F_{n-1}(u)]_{q^{-2}}\Big]_{q^2}=0\,,$$

$$\underset{\nu_1,\nu_2}{\text{Sym}}\Big[E_n(\nu_1),[E_n(\nu_2),E_{n-1}(u)]_{q^2}\Big]_{q^{-2}}=0\,,$$

$$\underset{\nu_1,\nu_2,\nu_3}{\text{Sym}}\Big[F_{n-1}(\nu_1),\Big[F_{n-1}(\nu_2),[F_{n-1}(\nu_3),F_n(u)]_{q^{-2}}\Big]_{q^2}\Big]=0\,,$$

$$\underset{\nu_1,\nu_2,\nu_3}{\text{Sym}}\Big[E_{n-1}(\nu_1),\Big[E_{n-1}(\nu_2),[E_{n-1}(\nu_3),E_n(u)]_{q^2}\Big]_{q^{-2}}\Big]=0\,.$$

## 7 Currents and the projections

Currents and the Gauss coordinates can be related through projections $P_f^\pm$ and $P_e^\pm$ onto sub-algebras $U_f^\pm$ and $U_e^\pm$ acting on the ordered products of the currents. Rigorous definitions of these projections depend on the type of the cycling ordering of the Cartan-Weyl generators (see [3] for detailed exposition of the properties of the projections for the ordering (4.41)).

Denote by $\overline{U}_f$ [20] an extension of the algebra $U_f = U_f^- \cup U_f^+ \cup U_k^+$ formed by linear combinations of series, given as infinite sums of monomials $a_{i_1}[n_1]\cdots a_{i_k}[n_k]$ with $n_1 \le \cdots \le n_k$, and $n_1 + ... + n_k$ fixed, where $a_{i_l}[n_l]$ is either $F_{i_l}[n_l]$ or $k_{i_l}^+[n_l]$. Analogously, denote by $\overline{U}_e$ an extension of the algebra $U_e = U_e^- \cup U_e^+ \cup U_k^-$ formed by linear combinations of series, given as infinite sums of monomials $a_{i_1}[n_1]\cdots a_{i_k}[n_k]$ with $n_1 \ge \cdots \ge n_k$, and $n_1 + ... + n_k$ fixed, where $a_{i_l}[n_l]$ is either $E_{i_l}[n_l]$ or $k_{i_l}^-[n_l]$.

It was proved in [3, 9, 20] that the ordered products of the simple roots currents $F_{j-1}(u)\cdots F_i(u)$ and $E_i(u)\cdots E_{j-1}(u)$ in the algebra $\tilde{U}_q(A_{N-1}^{(1)})$ are well defined and belong to $\overline{U}_f$ and $\overline{U}_e$ respectively. Moreover, the actions of the projections $P_f^\pm$ and $P_e^\pm$ onto these elements are well defined.

The action of the projections $P_f^\pm$ onto product of the currents can be defined as follows. In order to calculate projections from such product one has to substitute each current by the difference of the corresponding Gauss coordinates (6.64) and using the commutation relations between them order the product of the currents $F_i(u)$ in a way that all negative Gauss coordinates $F_{j,i}^-(u)$ will be on the left of all positive Gauss coordinates $F_{k,l}^+(\nu)$. Then application of the projection $P_f^+$ to this product of the currents is removing all the terms which have at least one negative Gauss coordinate on the left. Analogously, application of the projection $P_f^-$ is removing all the terms which have at least one positive Gauss coordinate on the right. The action of the projections $P_e^\pm$ onto product of the currents $E_i(u)$ is defined analogously, but the ordering of the Gauss coordinates is inverse: all positive Gauss coordinates $E_{i,j}^+(u)$ should be placed on the left of all negative coordinates $E_{l,k}^-(\nu)$ using the commutation relations between them and according to the ordering (4.41).

For $U_q(B_n^{(1)})$, $U_q(C_n^{(1)})$, $U_q(D_n^{(1)})$ and $U_q(A_{N-1}^{(2)})$ we introduced currents $F_i(u)$ and $E_i(u)$, $1 \le i \le n$ by the formulas (6.64) and (6.65). Using relations (5.62) we define dependent currents $F_i(u)$ and $E_i(u)$, $n' \le i \le N-1$

$$F_i(u)=-F_{N-i}(q^{2(i-N)}\xi^{-1}u),\quad E_i(u)=-E_{N-i}(q^{2(i-N)}\xi^{-1}u)\,.\tag{7.71}$$

For the algebras $U_q(B_n^{(1)})$ and $U_q(A_{2n}^{(2)})$ and according to (5.63) we introduce additional dependent currents

$$F_{n+1}(u)=-q^{1/2}\,F_n(q^{-2n}\xi^{-1}u),\quad E_{n+1}(u)=-q^{-1/2}\,E_n(q^{-2n}\xi^{-1}u)\,.\tag{7.72}$$

For $1 \leq i < j \leq N$ one can define the elements $\mathcal{F}_{j,i}(u)$ and $\mathcal{E}_{i,j}(u)$ from the completed subalgebras $\overline{U}_f$ and $\overline{U}_e$

$$
\begin{aligned}
\mathcal{F}_{j,i}(u) &= F_{j-1}(u)F_{j-2}(u)\cdots F_{i+1}(u)F_i(u)\,, \\
\mathcal{E}_{i,j}(u) &= E_i(u)E_{i+1}(u)\cdots E_{j-2}(u)E_{j-1}(u)\,,
\end{aligned}
\tag{7.73}
$$

for $U_q(B_n^{(1)})$, $U_q(C_n^{(1)})$, $U_q(A_{N-1}^{(2)})$ and

$$
\mathcal{F}_{j,i}(u) =
\begin{cases}
F_{j-1}(u)F_{j-2}(u)\cdots F_{i+1}(u)F_i(u), & i < j \leq n \quad \text{or} \quad n+1 \leq i < j\,, \\
0, & i = n,\ j = n+1\,, \\
F_n(u)F_{n-2}(u)\cdots F_i(u), & i \leq n-1,\ j = n+1\,, \\
-F_{j-1}(u)\cdots F_{n+2}(u)F_n(u), & i = n,\ j \geq n+2\,, \\
F_{j-1}(u)\cdots F_{n+2}(u)F_{n+1}(u)F_n(u)F_{n-2}(u)\cdots F_i(u), & i < n < j-1\,,
\end{cases}
$$

$$
\mathcal{E}_{i,j}(u) =
\begin{cases}
E_i(u)E_{i+1}(u)\cdots E_{j-2}(u)E_{j-1}(u), & i < j \leq n \quad \text{or} \quad n+1 \leq i < j\,, \\
0, & i = n,\ j = n+1\,, \\
E_i(u)\cdots E_{n-2}(u)E_n(u), & i \leq n-1,\ j = n+1\,, \\
-E_n(u)E_{n+2}(u)\cdots E_{j-1}(u), & i = n,\ j \geq n+2\,, \\
E_i(u)\cdots E_{n-2}(u)E_n(u)E_{n+1}(u)E_{n+2}(u)\cdots E_{j-1}(u), & i < n < j-1\,,
\end{cases}
$$

for $U_q(D_n^{(1)})$. To define composed currents for $U_q(D_n^{(1)})$ one can use commutativity $[F_{n-1}(u), F_n(v)] = 0$, $[E_{n-1}(u), E_n(v)] = 0$ and $F_{n+1}(u) = -F_{n-1}(u)$, $E_{n+1}(u) = -E_{n-1}(u)$ (see (7.71)).

We call elements $\mathcal{F}_{j,i}(u)$ and $\mathcal{E}_{i,j}(u)$ *the composed currents*. For the algebra $U_q(A_{N-1}^{(1)})$ these currents were investigated in [3, 9]. It was shown there that the analytical properties of the products of the composed currents considered in the category of the highest weight representations are equivalent to the Serre relations for the simple root currents.

Using result of [3] that action of the projections $P_f^\pm$ and $P_e^\pm$ can be prolonged to the extensions $\overline{U}_f$ and $\overline{U}_e$ respectively we formulate following

**Proposition 7.1.** *There are relations between Gauss coordinates of the fundamental L-operators and projections of the composed currents in the algebra $U_q(\tilde{\mathfrak{g}})$*

$$
\begin{aligned}
P_f^+\big(\mathcal{F}_{j,i}(u)\big) &= \mathrm{F}_{j,i}^+(u), & P_f^-\big(\mathcal{F}_{j,i}(u)\big) &= \tilde{\mathrm{F}}_{j,i}^-(u), \\
P_e^+\big(\mathcal{E}_{i,j}(u)\big) &= \mathrm{E}_{i,j}^+(u), & P_e^-\big(\mathcal{E}_{i,j}(u)\big) &= \tilde{\mathrm{E}}_{i,j}^-(u).
\end{aligned}
\tag{7.74}
$$

Proof of this Proposition will be given in Appendix D simultaneously for all algebras $U_q(\tilde{\mathfrak{g}})$ and is based on induction over rank $n$ of these algebras. The base of induction is a verification of (7.74) for all algebras $U_q(\tilde{\mathfrak{g}})$ of small ranks performed in the Appendices C.1, C.2, C.3, C.4 and C.5. In particular, formulas (C.110) and (C.111) are base of induction to prove Proposition 7.1 in case of the algebra $U_q(D_n^{(1)})$.

## 8 Conclusion

In this paper we investigate quantum loop algebras for all classical series (except $U_q(D_n^{(2)})$) associated to the quantum R-matrices found in [1]. Results obtained in this paper can be used for investigation of the space of states of the quantum integrable models with the different symmetries of the high rank. This investigation can be performed in the framework of the

approach to integrable models proposed and developed in [3, 9, 21]. In this method the states of integrable models are expressed through current generators of the quantum loop algebras. To investigate different physical quantities in such models such as scalar products of the states and form-factors of the local operators it is not necessary to have explicit form of the states in terms of the current generators. Usually, it is sufficient to get the action of monodromy matrix entries onto these states. This approach was called *zero modes method* and was already used in [22–24] to investigate the space of states in quantum integrable models related to Yangian doubles and rational $\mathfrak{g}$-invariant R-matrices. Using results of the present paper we plan to develop this method for the integrable models associated with $U_q(\mathfrak{g})$-invariant R-matrices.

**Funding information.** The study has been funded within the framework of the HSE University Basic Research Program.

# A Proofs of the Lemmas 5.2 and 5.3

Recall that we denote by $|i\rangle$ and $\langle j|$, $1 \le i, j \le N$ sets of orthonormal vectors in $\mathbb{C}^N$ with pairing $\mathbb{C}^N \otimes \mathbb{C}^N \to \mathbb{C}$: $\langle i|j\rangle = \delta_{ij}$. Consider R-matrix (2.9) for $v = q^2 u$: $R(1, q^2) = \mathbb{R}(1, q^2) + Q(1, q^2)$, where

$$\mathbb{R}(1, q^2) = \sum_{i \ne j}^N e_{ii} \otimes e_{jj} - q^{-1} \sum_{i<j}^N e_{ij} \otimes e_{ji} - q \sum_{i<j}^N e_{ji} \otimes e_{ij}$$

and calculate

$$R(1, q^2)|\ell, \ell\rangle = 0 \quad \langle \ell, \ell|R(1, q^2) = 0 \quad \text{for} \quad 1 \le \ell \le N \quad \text{and} \quad \ell \ne \ell', \tag{A.75}$$

and

$$\begin{aligned} R(1, q^2)|1, \ell\rangle &= |1, \ell\rangle - q|\ell, 1\rangle, \quad R(1, q^2)|\ell, 1\rangle = |\ell, 1\rangle - q^{-1}|1, \ell\rangle, \\ \langle \ell, 1|R(1, q^2) &= \langle \ell, 1| - q\langle 1, \ell|, \quad \langle 1, \ell|R(1, q^2) = \langle 1, \ell| - q^{-1}\langle \ell, 1|, \end{aligned} \tag{A.76}$$

for $1 < \ell < N$. Equation (A.76) implies

$$R(1, q^2)\big(|1, \ell\rangle + q|\ell, 1\rangle\big) = 0 \quad \text{for} \quad 1 < \ell < N. \tag{A.77}$$

Here $|i, j\rangle = |i\rangle \otimes |j\rangle$ and $\langle i, j| = \langle i| \otimes \langle j|$ are vectors from $(\mathbb{C}^N)^{\otimes 2}$.

Consider commutation relation (3.27) at $v = q^2 u$

$$R_{12}(1, q^2)\, L^{(1)}(u)L^{(2)}(q^2 u) = L^{(2)}(q^2 u)L^{(1)}(u)\, R_{12}(1, q^2). \tag{A.78}$$

Using (3.27) one can obtain the commutation relations between L-operators $L(u)$ and $\mathbb{L}(v)$

$$R_{12}(u, v)R_{13}(u, q^2 v)\, L^{(1)}(u)\, \mathbb{L}^{(2,3)}(v) = \mathbb{L}^{(2,3)}(v)\, L^{(1)}(u)\, R_{13}(u, q^2 v)R_{12}(u, v), \tag{A.79}$$

where Yang-Baxter equation (2.14)

$$R_{12}(u, v) \cdot R_{13}(u, q^2 v) \cdot R_{23}(1, q^2) = R_{23}(1, q^2) \cdot R_{13}(u, q^2 v) \cdot R_{12}(u, v)$$

is used.

Analogously, one can obtain the commutation relations between L-operators $\mathbb{L}(u)$ and $\mathbb{L}(v)$

$$\begin{aligned} R_{23}(q^2 u, v)R_{13}(u, v)R_{24}(u, v)R_{14}(u, q^2 v)\, \mathbb{L}^{(1,2)}(u)\, \mathbb{L}^{(3,4)}(v) \\ = \mathbb{L}^{(3,4)}(v)\, \mathbb{L}^{(1,2)}(u)\, R_{14}(u, q^2 v)R_{24}(u, v)R_{13}(u, v)R_{23}(q^2 u, v). \end{aligned} \tag{A.80}$$

To prove Lemma 5.2 one has to obtain two equalities for $1 < \ell < N$

$$R_{23}(1,q^2)R_{13}(u,q^2v)R_{12}(u,v)|1,\ell,1\rangle = f(u,v)R_{23}(1,q^2)|1,\ell,1\rangle \qquad (A.81)$$

and

$$\langle 1,\ell,1|R_{12}(u,v)R_{13}(u,q^2v)R_{23}(1,q^2) = f(u,v)\langle 1,\ell,1|R_{23}(1,q^2). \qquad (A.82)$$

One can verify equality (A.81)

$$
\begin{aligned}
&R_{23}(1,q^2)R_{13}(u,q^2v)R_{12}(u,v)|1,\ell,1\rangle \\
&= R_{23}(1,q^2)R_{13}(u,q^2v)\Big(|1,\ell,1\rangle + \mathsf{p}_{\ell 1}(u,v)|\ell,1,1\rangle\Big) \\
&= R_{23}(1,q^2)\Big((1+\mathsf{p}_{11}(u,q^2v))|1,\ell,1\rangle + \mathsf{p}_{1\ell}(u,q^2v)\mathsf{p}_{\ell 1}(u,v)|1,1,\ell\rangle\Big) \\
&= \Big(1+\mathsf{p}_{11}(u,q^2v)-q\,\mathsf{p}_{1\ell}(u,q^2v)\mathsf{p}_{\ell 1}(u,v)\Big)R_{23}(1,q^2)|1,\ell,1\rangle \\
&= f(u,v)R_{23}(1,q^2)|1,\ell,1\rangle,
\end{aligned}
$$

where identity

$$f(u,q^2v)-q\,g(u,q^2v)\tilde{g}(u,v) = f(u,v)$$

and equation (A.77) were used. Equality (A.82) can be checked analogously.

Multiplying equality (A.79) from the left by the vector $\langle 1,i,1|$ and from the right by the vector $|1,j,1\rangle$ for $1 < i,j < N$ and using (A.81) and (A.82) one obtains

$$\langle 1,i,1|L^{(1)}(u)\mathbb{L}^{(2,3)}(v)|1,j,1\rangle = \langle 1,i,1|\mathbb{L}^{(2,3)}(v)L^{(1)}(u)|1,j,1\rangle,$$

which implies the statement of the Lemma (5.2). $\qquad\square$

To prove equality (5.49) of the Lemma 5.3 one can present its left hand side

$$R^n_{12}(1,q^2)R^n_{34}(1,q^2)R^n_{14}(u,q^2v)R^n_{13}(u,v)|i,1,j,1\rangle,$$

as sum of two terms using (2.9): $R^n_{13}(u,v) = \mathbb{R}^n_{13}(u,v) + \mathbb{Q}^n_{13}(u,v)$. First term is equal to

$$
\begin{aligned}
&R^n_{12}(1,q^2)R^n_{34}(1,q^2)R^n_{14}(u,q^2v)\mathbb{R}^n_{13}(u,v)|i,1,j,1\rangle \\
&= R^n_{12}(1,q^2)R^n_{34}(1,q^2)\mathbb{R}^n_{13}(u,v)|i,1,j,1\rangle \qquad (A.83) \\
&= R^n_{12}(1,q^2)R^n_{34}(1,q^2)\mathbb{R}^{n-1}_{13}(u,v)|i,1,j,1\rangle,
\end{aligned}
$$

since $1 < i,j < N$. Indeed, the action of $R^n_{14}(u,q^2v)\mathbb{R}^n_{13}(u,v)$ onto vector $|i,1,j,1\rangle$ is

$$|i,1,j,1\rangle + \mathsf{p}_{ji}(u,v)|j,1,i,1\rangle + \mathsf{p}_{1i}(u,q^2v)|1,1,j,i\rangle + \mathsf{p}_{ji}(u,v)\mathsf{p}_{1j}(u,q^2v)|1,1,i,j\rangle$$

and last two terms are annihilated by the actions of $R_{12}(1,q^2)$ due to (A.75).

Now consider the second term

$$R^n_{12}(1,q^2)R^n_{34}(1,q^2)R^n_{14}(u,q^2v)\mathbb{Q}^n_{13}(u,v)|i,1,j,1\rangle,$$

where by definition (2.5)

$$\mathbb{Q}^n_{13}(u,v)|i,1,j,1\rangle = \delta_{ij'}\sum_{\ell=1}^{N}\mathsf{q}_{\ell j}(u,v)|\ell',1,\ell,1\rangle.$$

Calculating the action

$$R^n_{14}(u,q^2v)|\ell',1,\ell,1\rangle = |\ell',1,\ell,1\rangle + \mathsf{p}_{1\ell'}(u,q^2v)|1,1,\ell,\ell'\rangle + \delta_{1\ell}\sum_{m=1}^{N}\mathsf{q}_{m1}(u,q^2v)|m',1,1,m\rangle$$

one can observe that second term in the right hand side drops out due to the action $R_{12}^n(1, q^2)$ and (A.75) and by the same reasons the sum over $m$ reduces to the sum for $1 < m < N$. Finally, one gets

$$
\begin{aligned}
&R_{12}^n(1, q^2) R_{34}^n(1, q^2) R_{14}^n(u, q^2 v) \mathbb{Q}_{13}^n(u, v) |i, 1, j, 1\rangle \\
&= \delta_{ij'} R_{12}^n(1, q^2) R_{34}^n(1, q^2) \sum_{\ell=2}^{N-1} \Big( q_{\ell j}(u, v) |\ell', 1, \ell, 1\rangle + q_{1j}(u, v) q_{\ell 1}(u, q^2 v) |\ell', 1, 1, \ell\rangle \Big) \\
&= \delta_{ij'} R_{12}^n(1, q^2) R_{34}^n(1, q^2) \sum_{\ell=2}^{N-1} \Big( q_{\ell j}(u, v|\xi) - q\, q_{1j}(u, v|\xi) q_{\ell 1}(u, q^2 v|\xi) \Big) |\ell', 1, \ell, 1\rangle,
\end{aligned}
\tag{A.84}
$$

where in the last line of (A.84) we used (A.77) for $R_{34}^n(1, q^2)$ and write explicitly dependence of the functions $q_{ij}(u, v|\xi)$ given by (2.6) on parameter $\xi$.

One can check that for all algebras $\tilde{\mathfrak{g}} = B_n^{(1)}$, $C_n^{(1)}$, $D_n^{(1)}$ and $A_{N-1}^{(2)}$ and corresponding parameters $\xi$ given by the table (2.1) following identity is valid

$$
q_{\ell j}(u, v|\xi) - q\, q_{1j}(u, v|\xi) q_{\ell 1}(u, q^2 v|\xi) = q_{\ell j}(u, v|q^2 \xi).
$$

Since multiplication of the parameter $\xi$ by $q^2$ means the change of the rank $n \to n-1$ for all algebras $U_q(\tilde{\mathfrak{g}})$ (see table (2.1)) one concludes that

$$
\begin{aligned}
&R_{12}^n(1, q^2) R_{34}^n(1, q^2) R_{14}^n(u, q^2 v) \mathbb{Q}_{13}^n(u, v) |i, 1, j, 1\rangle \\
&= R_{12}^n(1, q^2) R_{34}^n(1, q^2) \mathbb{Q}_{13}^{n-1}(u, v) |i, 1, j, 1\rangle.
\end{aligned}
\tag{A.85}
$$

Summing (A.83) and (A.85) we obtain (5.49). Equality (5.50) can be proved analogously. This concludes the proof of the Lemma 5.3. □

# B  Proof of Proposition 5.5

Equalities (3.34) and (4.40) imply that

$$
L_{i,j}(u) = \varepsilon_i \varepsilon_j\, q^{\bar{i}-\bar{j}} \sum_{\ell \leq \min(i,j)} \tilde{E}_{i',\ell'}(\xi u)\, k_{\ell'}(\xi u)^{-1}\, \tilde{F}_{\ell',j'}(\xi u).
\tag{B.86}
$$

Comparing these expressions for the matrix entries of the fundamental L-operator with (5.54) proves equations (5.58), (5.59) and (5.60) for $1 \leq i < j \leq N$ and $1 \leq \ell \leq N$. Introduce matrix entries $\bar{M}_{i,j}(u)$ for the algebra $U_q^{n-1}(\tilde{\mathfrak{g}})$ by the equality

$$
L_{i,j}(u) = \bar{M}_{i,j}(u) + \bar{E}_{1,i}(u) \bar{k}_1(u) \bar{F}_{j,1}(u) = \bar{M}_{i,j}(u) + L_{i,1}(u) L_{1,1}(u)^{-1} L_{1,j}(u).
\tag{B.87}
$$

Entries $\bar{M}_{i,j}(u)$ for $1 < i, j < N$ has Gauss decomposition

$$
\begin{aligned}
\bar{M}_{i,j}(u) &= \sum_{2 \leq \ell \leq \min(i,j)} \bar{E}_{\ell,i}(q^{-2(\ell-1)} u)\, \bar{k}_\ell(q^{-2(\ell-1)} u)\, \bar{F}_{j,\ell}(q^{-2(\ell-1)} u) \\
&= L_{i,j}(u) - L_{i,1}(u) L_{1,1}(u)^{-1} L_{1,j}(u).
\end{aligned}
\tag{B.88}
$$

Calculating matrix elements of the equality (A.78) between vectors $\langle 1, 1|$ and $|1, j\rangle$ using (A.75) and (A.76) one obtains

$$
L_{1,j}(q^2 u) L_{1,1}(u) = q\, L_{1,1}(q^2 u) L_{1,j}(u)
\tag{B.89}
$$

and matrix entries (B.88) can be written in the form

$$\bar{M}_{i,j}(u) = \Big(L_{i,j}(u)L_{1,1}(q^{-2}u) - q\, L_{i,1}(u)L_{1,j}(q^{-2}u)\Big)L_{1,1}(q^{-2}u)^{-1}\,. \tag{B.90}$$

One can prove commutativity

$$\bar{M}_{i,j}(u)\, L_{1,1}(v) = L_{1,1}(v)\, \bar{M}_{i,j}(u)\,, \quad 1 < i,j < N\,,$$

in the same way as Lemma 5.2 was proved.

Multiplying (A.78) from the left and from the right by the vectors $\langle 1,i|$ and $|j,1\rangle$ for $1 < i,j < N$ and using (A.76) one gets

$$L_{i,j}(u)L_{1,1}(q^2u) - q\, L_{1,j}(u)L_{i,1}(q^2u) = L_{i,j}(q^2u)L_{1,1}(u) - q\, L_{i,1}(q^2u)L_{1,j}(u)$$

or due to (5.48) and (B.90)

$$\bar{M}_{i,j}(q^2u)L_{1,1}(q^2u)^{-1} = M_{i,j}(u)L_{1,1}(u)^{-1} \quad \text{for} \quad 1 < i,j < N\,. \tag{B.91}$$

We prove only (5.55) and (5.57). Equality (5.56) can be proved analogously. Comparing (4.37), (5.54) and (B.86) for the matrix entry $L_{1,1}(u)$ one concludes that

$$\bar{k}_1(u) = k_1(u) = k_N(\xi u)^{-1}\,. \tag{B.92}$$

Using $L_{1,j}(u) = F_{j,1}(u)k_1(u) = \bar{k}_1(u)\bar{F}_{j,1}(u)$ equality (B.89) can be rewritten in the form

$$\bar{F}_{j,1}(u) = qF_{j,1}(q^{-2}u) = k_1(u)^{-1}F_{j,1}(u)k_1(u)\,. \tag{B.93}$$

This yields

$$\bar{F}_{j,1}(u) = qF_{j,1}(q^{-2}u) = q^{\bar{1}-\bar{j}}\varepsilon_1\varepsilon_j\tilde{F}_{N,j'}(\xi u) \quad \text{for} \quad 1 < j < N \tag{B.94}$$

and (5.58) yields

$$\bar{F}_{N,1}(u) = q^{\bar{1}-\bar{N}}\varepsilon_1\varepsilon_N\tilde{F}_{N,1}(\xi u)\,. \tag{B.95}$$

Consider (B.91) for $i = j = 2$. It yields

$$\bar{k}_2(u) = k_2(u)\, k_1(q^2u)k_1(u)^{-1} = k_{N-1}(q^2\xi u)^{-1}\,, \tag{B.96}$$

where the second equality follows from (5.60). Then consider (B.91) for $i = 2$ and $2 < j < N-1$ to obtain

$$\bar{M}_{2,j}(u) = \bar{k}_2(q^{-2}u)\bar{F}_{j,2}(q^{-2}u) = k_1(u)k_1(q^{-2}u)^{-1}\, F_{j,2}(q^{-2}u)k_2(q^{-2}u)\,,$$

which can be presented as

$$\bar{F}_{j,2}(u) = k_2(u)^{-1}F_{j,2}(u)k_2(u)\,.$$

Recall now that according to the theorem 5.1 L-operator $M(u)$ satisfy commutation relations (5.46) for the algebra $U_q^{n-1}(\tilde{\mathfrak{g}})$ and we can apply analysis as above to have $k_2(u)^{-1}F_{j,2}(u)k_2(u) = q\, F_{j,2}(q^{-2}u)$ (compare with (B.93)) and

$$\bar{F}_{j,2}(u) = q\, F_{j,2}(q^{-2}u) = q^{\bar{2}-\bar{j}}\varepsilon_2\varepsilon_j\tilde{F}_{N-1,j'}(q^2\xi u) \quad \text{for} \quad 2 < j < N-1\,. \tag{B.97}$$

The second equality in (B.97) follows from (B.86) as well as

$$\bar{F}_{j,2}(u) = q^{\bar{2}-\bar{j}}\varepsilon_2\varepsilon_j\tilde{F}_{N-1,j'}(q^2\xi u)\,, \quad j = N-1, N\,. \tag{B.98}$$

Note that equalities (B.97), (B.98) and second equality in (B.96) for the Gauss coordinates of the embedded algebra $U_q^{n-1}(\tilde{\mathfrak{g}})$ repeated the equalities (B.94), (B.95) and (B.92) respectively with the only difference that parameter $\xi$ is replaced by the parameter $q^2\xi$. According to dependence of $\xi$ on the rank $n$ of the algebra $\tilde{\mathfrak{g}}$ this replacement is equivalent to change of the rank $n \to n-1$.

Continuing embedding process and repeating these arguments for the Gauss coordinates $\bar{E}_{i,j}(u)$ one proves Proposition 5.5. $\qquad\square$

# C Algebra $U_q(\tilde{\mathfrak{g}})$ for small ranks

In this Appendix we obtain the commutation relations for the currents $F_n(u)$ and $E_n(u)$ for each of the algebra $U_q(\tilde{\mathfrak{g}})$ of the small rank. Here we will introduce different rational functions denoting them by the same notations valid inside of each subsection. Hope that this will not lead to misunderstanding.

## C.1 Algebras $U_q(B_1^{(1)})$ and $U_q(B_2^{(1)})$

In order to find commutation relations of the special currents in case of the algebra $U_q(B_n^{(1)})$ we first perform investigation of the simplest nontrivial example of the algebra $U_q(B_1^{(1)})$ as it was done in the paper [23]. In this algebra the algebraically independent series of generators are $k_1^\pm(u)$, $F_{2,1}^\pm(u)$ and $E_{1,2}^\pm(u)$ and algebraically dependent generating series are $k_\ell^\pm(u)$, $\ell = 2, 3$ and

$$F_{3,2}^\pm(u) = -q^{1/2} F_{2,1}^\pm(q^{-1}u), \quad E_{2,3}^\pm(u) = -q^{-1/2} E_{1,2}^\pm(q^{-1}u),$$

$$F_{3,1}^\pm(v) = -\frac{\sqrt{q}}{1+q} F_{2,1}^\pm(v)^2, \quad E_{1,3}^\pm(v) = -\frac{\sqrt{q}}{1+q} E_{1,2}^\pm(v)^2.$$

The modes of $k_\ell^\pm(u)$, $\ell = 2, 3$ are defined by the relations

$$k_3^\pm(u) = k_1^\pm(qu)^{-1}, \quad k_1^\pm(u) = k_2^\pm(qu) \, k_2^\pm(u) \, k_1^\pm(q^2u).$$

The commutation relations between Gauss coordinates for the algebra $U(B_1^{(1)})$ are

$$k_1(u)F_{2,1}(v)k_1(u)^{-1} = f(v,u)F_{2,1}(v) - g(v,u)F_{2,1}(u),$$
$$k_1(u)^{-1}E_{1,2}(v)k_1(u) = f(v,u)E_{1,2}(v) - \tilde{g}(v,u)E_{1,2}(u), \quad \text{(C.99)}$$
$$[E_{1,2}(v), F_{2,1}(u)] = g(u,v)\left(k_2(u)k_1(u)^{-1} - k_2(v)k_1(v)^{-1}\right),$$

$$k_2(u)F_{2,1}(v)k_2(u)^{-1} = f(v,u)f(q^{-1}u,v)F_{2,1}(v) + g(v,u)F_{2,1}(u) + \tilde{g}(q^{-1}u,v)F_{2,1}(q^{-1}u),$$
$$k_2(u)^{-1}E_{1,2}(v)k_2(u) = f(v,u)f(q^{-1}u,v)E_{1,2}(v) + \tilde{g}(v,u)E_{1,2}(u) + g(q^{-1}u,v)E_{1,2}(q^{-1}u),$$
$$\text{(C.100)}$$

$$F_{2,1}(u)F_{2,1}(v) = f(u,qv)F_{2,1}(v)F_{2,1}(u) + \frac{g(qv,u)}{1+q}F_{2,1}(u)^2 + \frac{q\tilde{g}(qv,u)}{1+q}F_{2,1}(v)^2,$$
$$E_{1,2}(u)E_{1,2}(v) = f(v,qu)E_{1,2}(v)E_{1,2}(u) + \frac{g(qu,v)}{1+q}E_{1,2}(u)^2 + \frac{q\tilde{g}(qu,v)}{1+q}E_{1,2}(v)^2. \quad \text{(C.101)}$$

Restoring upper indices $\pm$ in (C.101) at $u = q^{-1}v$

$$F_{2,1}^+(q^{-1}v)F_{2,1}^\pm(v) = \frac{1}{1+q^{-1}}F_{2,1}^+(q^{-1}v)^2 + \frac{1}{1+q}F_{2,1}^\pm(v)^2$$

and subtracting one equality from another one gets

$$F_{2,1}^+(q^{-1}v)F_1(v) = \frac{1}{1+q}F_{2,1}^+(v)^2 - \frac{1}{1+q}F_{2,1}^-(v)^2. \quad \text{(C.102)}$$

Using (C.102) one can calculate the projection $P_f^+(F_2(u)F_1(u))$ onto subalgebra $U_f^+$ assuming that Gauss coordinates in the product of the currents $F_2(u)F_1(u)$ are ordered according to the order (4.41). We obtain

$$P_f^+(F_2(u)F_1(u)) = -\sqrt{q}\, P_f^+\left(F_{2,1}^+(q^{-1}u)F_1(u)\right) = -\frac{\sqrt{q}}{1+q}F_{2,1}^+(v)^2 = F_{3,1}^+(u).$$

This relation together with analogous formulas for the projections $P_f^-(F_2(u)F_1(u))$ and $P_e^\pm(E_1(u)E_2(u))$ are base of the induction proof of the Proposition 7.1 which explains the relation between Gauss coordinates and projection of the currents for the algebra $U_q(B_n^{(1)})$.

Considering similar commutation relations for the algebra $U_q(B_2^{(1)})$ and using embedding theorem 5.1 one obtains besides commutation relations (C.99)–(C.101) for the Gauss coordinates $k_\ell^\pm(u)$, $\ell = 1, 2, 3$ with $F_{3,2}^\pm(u)$ and $E_{2,3}^\pm(u)$ also the commutation relations of these Gauss coordinates with $F_{2,1}^\pm(u)$ and $E_{1,2}^\pm(u)$

$$F_{2,1}(v)F_{3,2}(u) = f(u,v)F_{3,2}(u)F_{2,1}(v) + \tilde{g}(u,v)\Big(F_{3,1}(u) - F_{3,2}(u)F_{2,1}(u)\Big) - g(u,v)F_{3,1}(v),$$

$$E_{2,3}(u)E_{1,2}(v) = f(u,v)E_{1,2}(v)E_{2,3}(u) + g(u,v)\Big(E_{1,3}(u) - E_{1,2}(u)E_{2,3}(u)\Big) - \tilde{g}(u,v)E_{1,3}(v).$$

This information is sufficient to obtain for the algebra $U_q(B_n^{(1)})$ the commutation relations of the currents $F_n(u)$, $E_n(u)$, Gauss coordinates $k_\ell^\pm(u)$, $1 \le \ell \le n+1$ and the currents $F_i(u)$, $E_i(u)$, $1 \le i \le n-1$ given in section 6.1.

## C.2   Algebra $U_q(C_2^{(1)})$

Since algebra $U_q(C_1^{(1)})$ is not representative we start to consider first algebra $U_q(C_2^{(1)})$. In this algebra the algebraically independent generating series are $F_{\ell+1,\ell}^\pm(u)$, $E_{\ell,\ell+1}^\pm(u)$ and $k_\ell^\pm$ for $\ell = 1, 2$. Gauss coordinates $F_{2,1}^\pm(u)$, $E_{1,2}^\pm(u)$, $k_1^\pm(u)$ and $k_2^\pm(u)$ form the subalgebra in $U_q(C_2^{(1)})$ isomorphic to the algebra $\tilde{U}_q(A_1^{(1)})$ and we do not write explicitly commutation relations between them.

Introduce the rational functions relevant to the considered case

$$f(u,v) = \frac{q^2 u - q^{-2} v}{u - v}, \quad g(u,v) = \frac{(q^2 - q^{-2})u}{u - v}, \quad \tilde{g}(u,v) = \frac{(q^2 - q^{-2})v}{u - v}.$$

The rest commutation relations in $U_q(C_2^{(1)})$ can be written in the form

$$k_2(v)F_{3,2}(u)k_2(v)^{-1} = f(u,v)F_{3,2}(u) - g(u,v)F_{3,2}(v),$$

$$k_2(v)^{-1}E_{2,3}(u)k_2(v) = f(u,v)E_{2,3}(u) - \tilde{g}(u,v)E_{2,3}(v),$$

$$f(v,u)F_{3,2}(u)F_{3,2}(v) = f(u,v)F_{3,2}(v)F_{3,2}(u) + g(v,u)F_{3,2}(u)^2 - g(u,v)F_{3,2}(v)^2,$$

$$f(u,v)E_{2,3}(u)E_{2,3}(v) = f(v,u)E_{2,3}(v)E_{2,3}(u) + \tilde{g}(u,v)E_{2,3}(v)^2 - \tilde{g}(v,u)E_{2,3}(u)^2, \qquad \text{(C.103)}$$

$$F_{2,1}(v)F_{3,2}(u) = f(u,v)F_{3,2}(u)F_{2,1}(v) + \tilde{g}(u,v)\Big(F_{3,1}(u) - F_{3,2}(u)F_{2,1}(u)\Big) - g(u,v)F_{3,1}(v),$$

$$E_{2,3}(u)E_{1,2}(v) = f(u,v)E_{1,2}(v)E_{2,3}(u) + g(u,v)\Big(E_{1,3}(u) - E_{1,2}(u)E_{2,3}(u)\Big) - \tilde{g}(u,v)E_{1,3}(v),$$

$$[E_{2,3}(v), F_{3,2}(u)] = g(u,v)\Big(k_3(u)k_2(u)^{-1} - k_3(v)k_2(v)^{-1}\Big),$$

where diagonal Gauss coordinate $k_3(u)$ due to (5.57) and (5.60) is equal to

$$k_3(u) = k_2(q^4 u)^{-1} k_1^\pm(q^4 u) k_1^\pm(q^6 u)^{-1}.$$

These commutation relations allows to restore the full set of the commutation relations in terms of the currents for the algebra $U_q(C_n^{(1)})$ given in section 6.2.

To obtain the commutation relation (C.103) one has to use (3.28) for the values of the indices $\{i, j, k, l\} \to \{2, 3, 1, 2\}$ and $\{i, j, k, l\} \to \{2, 4, 1, 1\}$ which results to

$$f(u,v)F_{3,2}(u)F_{2,1}(v) = \frac{f(u,v)}{f(u,v)}F_{2,1}(v)F_{3,2}(u) + \frac{f(u,v)g(u,v)}{f(u,v)}F_{3,1}(v)$$

$$+ g(v,u)\Big(F_{3,1}(u) - F_{3,2}(u)F_{2,1}(u)\Big) + \frac{q^{-2}g(v,u)}{f(u,v)}F_{4,2}(u).$$

Considering the latter relation at $v = q^2 u$ we obtain

$$F_{4,2}(u) = q^2\Big(F_{3,1}(u) - F_{3,2}(u)F_{2,1}(u)\Big) \tag{C.104}$$

and (C.103). Now one can calculate the projection $P_f^+(F_3(u)F_2(u))$, where dependent current $F_3(u) = -F_1(q^4 u)$ is defined by (7.71) for $N = 4$ and $\xi = q^{-6}$. Restoring in (C.103) superscripts of the matrix entries and setting $v = q^4 u$ we obtain

$$F_{2,1}^+(q^4 u)F_{3,2}^\pm(u) = q^2\Big(F_{3,2}^\pm(u)F_{2,1}^\pm(u) - F_{3,1}^\pm(u)\Big) + q^{-2}F_{3,1}^+(q^4 u).$$

Calculating projection $P_f^+(F_3(u)F_2(u))$ onto $U_f^+$ according to the ordering (4.41) one gets

$$\begin{aligned}
P_f^+\left(F_3(u)F_2(u)\right) &= -P_f^+\left(F_1(q^4 u)F_2(u)\right) = -P_f^+\left(F_{2,1}^+(q^4 u)F_2(u)\right) \\
&= q^2\Big(F_{3,1}^+(u) - F_{3,2}^+(u)F_{2,1}^+(u)\Big) = F_{4,2}^+(u).
\end{aligned}$$

Analogously, one can prove that

$$P_f^+\left(F_2(u)F_1(u)\right) = F_{3,1}^+(u) \quad \text{and} \quad P_f^+\left(F_3(u)F_2(u)F_1(u)\right) = F_{4,1}^+(u).$$

These relations together with analogous relations for the currents $E_i(u)$ are base of the induction for the proof of the Proposition 7.1 in case of the algebra $U_q(C_n^{(1)})$.

## C.3   Algebra $U_q(D_2^{(1)})$

As above we start to consider algebra $U_q(D_n^{(1)})$ for small $n$. The case $n = 1$ is not representative and we begin with the case $n = 2$ to prove that $F_{3,2}^\pm(u) = E_{2,3}^\pm(u) = 0$.

Excluding term $L_{21}(v)L_{24}(u)$ from the commutation relation (3.28) with set of indices $\{i,j,k,l\} \to \{2,3,2,2\}$ and $\{i,j,k,l\} \to \{2,4,2,1\}$ we have

$$\begin{aligned}
&f(u,v)L_{2,3}(u)L_{2,2}(v) - f(v,u)L_{2,2}(v)L_{2,3}(u) \\
&\quad = f(v,u)\tilde{g}(v,u)L_{2,4}(v)L_{2,1}(u) - f(u,v)\tilde{g}(u,v)L_{2,4}(u)L_{2,1}(v).
\end{aligned} \tag{C.105}$$

This relation after setting $v = q^{-2}u$ and projecting onto subalgebras $U_f^\pm \cup U_k^\pm$ in the algebra $U_q(D_2^{(1)})$ yields the equality

$$F_{3,2}^\pm(u)k_2^\pm(u)k_2^\pm(q^{-2}u) = 0.$$

Since Gauss coordinates $k_2^\pm(u)$ are invertible it results that

$$F_{3,2}^\pm(u) = 0. \tag{C.106}$$

Analogously one can prove

$$E_{2,3}^\pm(u) = 0. \tag{C.107}$$

In order to find relations between Gauss coordinates $F_{j,1}(u)$, $j = 2,3,4$ one can consider the commutation relation (3.28) for the values of the indices $\{i,j,k,l\} \to \{1,3,1,2\}$ and $\{i,j,k,l\} \to \{1,4,1,1\}$. Excluding term $L_{11}(v)L_{14}(u)$ we have

$$\begin{aligned}
&f(u,v)L_{1,3}(u)L_{1,2}(v) = f(v,u)L_{1,2}(v)L_{1,3}(u) \\
&\quad + f(v,u)\tilde{g}(v,u)L_{1,4}(v)L_{1,1}(u) - f(u,v)\tilde{g}(u,v)L_{1,4}(u)L_{1,1}(v).
\end{aligned} \tag{C.108}$$

Using explicit expressions for the matrix entries $L_{1,j}(u)$ through Gauss coordinates (4.37), multiplying both equalities by the product of $k_1(u)^{-1}k_1(v)^{-1}$ and using the commutation relations

$$k_1(u)F_{j,1}(v)k_1(u)^{-1} = f(v,u)F_{j,1}(v) - g(v,u)F_{j,1}(u), \quad j = 2,3 \tag{C.109}$$

one can get from (C.108)

$$f(u,v)f(v,u)\Big(F_{2,1}(v)F_{3,1}(u)-F_{3,1}(u)F_{2,1}(v)\Big)=g(u,v)f(v,u)\Big(F_{4,1}(v)+F_{2,1}(v)F_{3,1}(v)\Big)$$
$$+g(v,u)f(u,v)\Big(F_{4,1}(u)+F_{3,1}(u)F_{2,1}(u)\Big).$$

Taking in this equality $u=q^2v$ and $u=q^{-2}v$ one can find the relations

$$F_{4,1}(u)=-F_{2,1}(u)F_{3,1}(u)=-F_{3,1}(u)F_{2,1}(u),$$
$$F_{2,1}(v)F_{3,1}(u)=F_{3,1}(u)F_{2,1}(v).$$
(C.110)

In the same way one can prove that

$$E_{1,4}(u)=-E_{1,2}(u)E_{1,3}(u)=-E_{1,3}(u)E_{1,2}(u),$$
$$E_{1,2}(v)E_{1,3}(u)=E_{1,3}(u)E_{1,2}(v).$$
(C.111)

Equalities (5.55), (5.56), (5.58) and (5.59) yields in this case

$$F_{4,5-j}(u)=-F_{j,1}(u),\quad E_{5-j,4}(u)=-E_{1,j}(u),\quad j=2,3.$$
(C.112)

Using (3.28) for $\{i,j,k,l\}\to\{2,2,1,3\}$ we can calculate

$$k_2(u)F_{3,1}(v)k_2(u)^{-1}=f(v\xi,u)F_{3,1}(v)-q\,\tilde{g}(u,v\xi)k_1(v)F_{3,1}(u)k_1(v)^{-1},$$

where we have used (C.106) and (C.112). Using now (C.109) and identities

$$f(v\xi,u)-q\,\tilde{g}(u,v\xi)g(u,v)=f(v,u),\quad q\,\tilde{g}(u,v\xi)f(u,v)=-g(v,u)$$

one can find that

$$k_2(u)F_{3,1}(v)k_2(v)^{-1}=f(v,u)F_{3,1}(v)-g(v,u)F_{3,1}(u).$$

Analogously one can obtain

$$k_2(u)^{-1}E_{1,3}(v)k_2(v)=f(v,u)E_{1,3}(v)-\tilde{g}(v,u)E_{1,3}(u).$$

Using (C.109) and analogous commutation relations for $E_{1,j}(u)$ one can calculate from the commutation relation (3.28) at $\{i,j,k,l\}\to\{1,3,3,1\}$ that

$$[E_{1,3}(v),F_{3,1}(u)]=g(u,v)\Big(k_3(u)k_1(u)^{-1}-k_3(v)k_1(v)^{-1}\Big).$$

The embedding theorem 5.1 and commutation relations between Gauss coordinates obtained for the algebra $U_q(D_2^{(1)})$ are sufficient to get full set of the commutation relations for the algebra $U_q(D_n^{(1)})$ in terms of the currents presented in the section 6.3.

## C.4  Algebra $U_q(A_2^{(2)})$

RLL realization of this algebra is given by the R-matrix (2.9) with $\xi=-q^{-1-2n}$ and $N=2n+1$. In the same way as we investigated the algebra $U_q(B_1^{(1)})$ we study first the algebra $U_q(A_{2n}^{(2)})$ in the simplest case $n=1$.

Introduce the functions

$$f(u,v)=\frac{q^{1/2}u+q^{-1/2}v}{u+v},\quad g(u,v)=\frac{(q^{1/2}+q^{-1/2})u}{u+v}.$$

The commutation relations for the Gauss coordinates in the algebra $U_q(A_2^{(2)})$ between $F_{2,1}(u)$, $k_1(u)$ and $E_{1,2}(u)$ are the same as for $U_q(B_1^{(1)})$ (see (C.99)). The rest relations are

$$k_2(u)F_{2,1}(v)k_2(u)^{-1} = f(u,v)\frac{f(v,u)}{f(u,v)}F_{2,1}(v) + g(v,u)F_{2,1}(u) + (1-q)\frac{g(v,u)}{f(u,v)}F_{2,1}(-qu),$$

$$k_2(u)^{-1}E_{1,2}(v)k_2(u) = f(u,v)\frac{f(v,u)}{f(u,v)}E_{1,2}(v) + \tilde{g}(v,u)E_{1,2}(u) + (q-1)\frac{g(u,v)}{f(u,v)}E_{1,2}(-qu)$$

and

$$\begin{aligned}
f(u,v)f(v,u)F_{2,1}(u)F_{2,1}(v) &= f(v,u)f(u,v)F_{2,1}(v)F_{2,1}(u) \\
&\quad + g(u,v)g(v,u)\left(F_{2,1}(u)^2 + F_{2,1}(v)^2\right) \\
&\quad + \frac{1}{q+1}[F_{2,1}[0], g(u,v)F_{2,1}(v) - g(v,u)F_{2,1}(u)]_q,
\end{aligned} \tag{C.113}$$

$$\begin{aligned}
f(v,u)f(u,v)E_{1,2}(u)E_{1,2}(v) &= f(u,v)f(v,u)E_{1,2}(v)E_{1,2}(u) \\
&\quad + g(v,u)g(u,v)\left(E_{1,2}(u)^2 + E_{1,2}(v)^2\right) \\
&\quad + \frac{1}{q^{-1}+1}[E_{1,2}[0], g(v,u)E_{1,2}(v) - g(u,v)E_{1,2}(u)]_{q^{-1}}.
\end{aligned} \tag{C.114}$$

To prove (C.113) one can use the commutation relation

$$\begin{aligned}
f(v,u)F_{2,1}^+(u)F_{2,1}^\pm(v) &= f(v,-qu)f(u,v)F_{2,1}^\pm(v)F_{2,1}^+(u) \\
&\quad + g(v,u)F_{2,1}^+(u)^2 + f(v,-qu)\tilde{g}(v,u)F_{2,1}^\pm(v)^2 \\
&\quad + q^{-1/2}g(v,-qu)F_{3,1}^+(u) + q^{-3/2}\tilde{g}(v,-qu)F_{3,1}^\pm(v).
\end{aligned} \tag{C.115}$$

Putting $v \to \infty$ in (C.115) one obtains

$$F_{3,1}^+(u) = -\sqrt{q}\, F_{2,1}^+(u)^2 - \frac{\sqrt{q}}{q-q^{-1}}[F_{2,1}[0], F_{2,1}^+(u)]_q. \tag{C.116}$$

Setting $u = -qv$ in (C.115) one finds

$$F_{2,1}^+(-qv)F_{2,1}^\pm(v) = (1-q^{-1})F_{2,1}^+(-qv)^2 - q^{-3/2}F_{3,1}^+(-qv) - q^{-1/2}F_{3,1}^\pm(v).$$

Using this equality one can calculate

$$P_f^+(F_2(v)F_1(v)) = -\sqrt{q}P_f^+(F_{2,1}^+(-qv)F_1(v)) = F_{3,1}^+(v), \tag{C.117}$$

where dependent current $F_2(v) = -\sqrt{q}F_1(-qv)$ is defined by (7.72) for $n=1$ and $\xi = -q^{-3}$. Relation (C.117) as well as analogous relation for the projections $P_e^\pm(E_1(u)E_2(u))$ are base of induction to prove Proposition 7.1 in case of the algebra $U_q(A_{2n}^{(2)})$.

## C.5 Algebras $U_q(A_1^{(2)})$ and $U_q(A_3^{(2)})$

Algebra $U_q(A_{2n-1}^{(2)})$ is defined by the R-matrix (2.9) with $\xi = -q^{-2n}$ and $N = 2n$. In this Appendix we investigate algebra $U_q(A_{2n-1}^{(2)})$ in two simplest cases $n=1$ and $n=2$.

For $n=1$ R-matrix (2.9) has the form

$$R(u,v) = \frac{u+v}{qu+q^{-1}v}\begin{pmatrix} f(u,v) & 0 & 0 & 0 \\ 0 & 1 & g(u,v) & 0 \\ 0 & g(u,v) & 1 & 0 \\ 0 & 0 & 0 & f(u,v) \end{pmatrix},$$

where rational functions $f(u, v)$ and $g(u, v)$ are defined as follows

$$f(u, v) = \frac{q^2 u^2 - q^{-2} v^2}{u^2 - v^2}, \qquad g(u, v) = \frac{(q^2 - q^{-2}) u v}{u^2 - v^2}.$$

Up to overall factor this matrix coincides with the symmetric form of the R-matrix for the algebra $U_{q^2}(A_1^{(1)})$. This results that the commutation relation for the Gauss coordinates $F_{2,1}(u)$, $E_{1,2}(u)$, $k_1(u)$ and $k_2(u)$ of the algebra $U_q(A_1^{(2)})$ can be written in the form

$$k_1(u) F_{2,1}(v) k_1(u)^{-1} = f(v, u) F_{2,1}(v) - g(v, u) F_{2,1}(u), \tag{C.118}$$

$$k_1(u)^{-1} E_{1,2}(v) k_1(u) = f(v, u) E_{1,2}(v) - g(v, u) E_{1,2}(u), \tag{C.119}$$

$$[E_{1,2}(u), F_{2,1}(v)] = g(v, u) \left( k_2(v) k_1(v)^{-1} - k_2(u) k_1(u)^{-1} \right), \tag{C.120}$$

$$k_2(u) F_{2,1}(v) k_2(u)^{-1} = f(u, v) F_{2,1}(v) - g(u, v) F_{2,1}(u), \tag{C.121}$$

$$k_2(u)^{-1} E_{1,2}(v) k_2(u) = f(u, v) E_{1,2}(v) - g(u, v) E_{1,2}(u), \tag{C.122}$$

and

$$f(v, u) F_{2,1}(u) F_{2,1}(v) - g(v, u) F_{2,1}(u)^2 = f(u, v) F_{2,1}(v) F_{2,1}(u) - g(u, v) F_{2,1}(v)^2, \tag{C.123}$$

$$f(u, v) E_{1,2}(u) E_{1,2}(v) - g(u, v) E_{1,2}(v)^2 = f(v, u) E_{1,2}(v) E_{1,2}(u) - g(v, u) E_{1,2}(u)^2. \tag{C.124}$$

The commutation relations (C.118)–(C.120) imply certain analytical properties of the Gauss coordinates $F_{2,1}(u)$, $E_{1,2}(u)$ and $k_2(u) k_1^{\pm}(u)^{-1}$. Indeed, setting in (C.118) and (C.119) $v = \pm q^{-2} u$ we obtain

$$\begin{aligned}
k_1(u)^{-1} F_{2,1}(u) k_1(u) &= \pm F_{2,1}(\pm q^{-2} u), \\
k_1(u) E_{1,2}(u) k_1(u)^{-1} &= \pm E_{1,2}(\pm q^{-2} u)
\end{aligned} \tag{C.125}$$

which imply

$$F_{2,1}(-u) = -F_{2,1}(u) \quad \text{and} \quad E_{1,2}(-u) = -E_{1,2}(u). \tag{C.126}$$

These equalities signify that Gauss coordinates $F_{2,1}^{\pm}(u)$ and $E_{1,2}^{\pm}(u)$ are series with respect of odd powers of the spectral parameters and equalities (C.125) are simplified to

$$\begin{aligned}
k_1(u)^{-1} F_{2,1}(u) k_1(u) &= F_{2,1}(q^{-2} u), \\
k_1(u) E_{1,2}(u) k_1(u)^{-1} &= E_{1,2}(q^{-2} u).
\end{aligned} \tag{C.127}$$

On the other hand, replacing $u$ by $-u$ in (C.120), using (C.126) and the fact that $g(-u, v) = -g(u, v)$ one obtains that

$$k_2(u) k_1(u)^{-1} = k_2(-u) k_1(-u)^{-1} \tag{C.128}$$

which signifies that the ration $k_2^{\pm}(u) k_1^{\pm}(u)^{-1}$ are series with respect to even powers of the spectral parameter. Moreover, equality (3.34) together with (C.127) yield in this case that $k_2^{\pm}(u) = k_1^{\pm}(-q^2 u)^{-1}$. Together with (C.128) it proves that in the algebra $U_q(A_1^{(2)})$ both diagonal Gauss coordinates $k_1^{\pm}(u)$ and $k_2^{\pm}(u)$ are series with respect to even powers of the spectral parameters.

In the case $n = 2$ and according to the embedding theorem 5.1 the Gauss coordinates $F_{2,1}(u)$, $E_{1,2}(u)$, $k_1(u)$ and $k_2(u)$ satisfy the commutation relations in the algebra $\tilde{U}_q(A_1^{(1)})$ while Gauss coordinates $F_{3,2}(u)$, $E_{2,3}(u)$, $k_2(u)$ and $k_3(u)$ satisfy the commutation relations (C.118)–(C.124). Commutation relations between these algebraically independent sets of the Gauss coordinates take the form

$$F_{2,1}(v)F_{3,2}(u) = f(u,v)F_{3,2}(u)F_{2,1}(v) - g(u,v)F_{3,1}(v)$$
$$+ g(u,v)\big(F_{3,1}(u) - F_{3,2}(u)F_{2,1}(u)\big) + \frac{1}{[2]_q}\frac{g(u,v)}{\tilde{g}(u,v)}\,[F_{2,1}[0], F_{3,2}(u)]_{q^{-2}}\,, \tag{C.129}$$

$$E_{2,3}(u)E_{1,2}(v) = f(u,v)E_{1,2}(v)E_{2,3}(u) - g(u,v)E_{1,3}(v)$$
$$+ g(u,v)\big(E_{1,3}(u) - E_{1,2}(u)E_{2,3}(u)\big) + \frac{1}{[2]_q}\frac{g(u,v)}{g(u,v)}\,[E_{2,3}(u), E_{1,2}[0]]_{q^2}\,,$$

where

$$[2]_q = q + q^{-1} = \frac{q^2 - q^{-2}}{q - q^{-1}}\,.$$

To prove equality (C.129) one can use an equality

$$F_{2,1}^+(v)F_{3,2}^\pm(u) = f(u,v)F_{3,2}^\pm(u)F_{2,1}^+(v) - g(u,v)F_{3,1}^+(v)$$
$$+ g(u,v)\frac{qu + q^{-1}v}{(q + q^{-1})u}\big(F_{3,1}^\pm(u) - F_{3,2}^\pm(u)F_{2,1}^\pm(u)\big) - \frac{(1 - q^{-2})v}{u + v}F_{4,2}^\pm(u) \tag{C.130}$$

which helps to calculate $P_f^+(F_3(u)F_2(u))$, where $F_3(u) = -F_1(-q^2 u)$ is a dependent current defined by (7.71) for $N = 4$ and $\xi = -q^{-4}$. Putting $v \to \infty$ in (C.130) one obtains

$$F_{4,2}^+(u) = F_{3,2}^+(u)F_{2,1}^+(u) - F_{3,1}^+(u) - \frac{q}{q - q^{-1}}[F_{2,1}[0], F_{3,2}^+(u)]_{q^{-2}}\,. \tag{C.131}$$

Setting in (C.130) $v = -q^2 u$ one gets

$$F_{4,3}^+(u)F_{3,2}^\pm(u) = F_{3,1}^+(-q^2 u) + F_{4,2}^\pm(u)$$

which implies

$$P_f^+\left(F_3(u)F_2(u)\right) = P_f^+\left(F_{4,3}^+(u)(F_{3,2}^+(u) - F_{3,2}^-(u))\right) = F_{4,2}^+(u)\,. \tag{C.132}$$

Analogously, considering equality (C.130) at $v = u$ one can prove that

$$P_f^+\left(F_2(u)F_1(u)\right) = P_f^+\left(F_{3,2}^+(u)(F_{2,1}^+(u) - F_{2,1}^-(u))\right) = F_{3,1}^+(u)\,. \tag{C.133}$$

Let us consider commutation relation (3.28) for the values of the indices $\{i,j,k,l\} \to \{2,4,1,2\}$ and $\{i,j,k,l\} \to \{2,4,1,1\}$ to obtain commutation of the Gauss coordinates

$$f(u,v)F_{4,2}^+(u)F_{2,1}^\pm(v) = \frac{1}{f(v\xi,u)}\,F_{2,1}^\pm(v)F_{4,2}^+(u)$$
$$+ \frac{g(u,v\xi)}{qf(v\xi,u)}\Big(F_{2,1}^\pm(v)F_{3,1}^\pm(v) + F_{2,1}^\pm(v)^2F_{3,2}^+(u)\Big) \tag{C.134}$$
$$+ g(u,v)F_{4,1}^\pm(v) - \tilde{g}(u,v)\Big(F_{4,1}^+(u) - F_{4,2}^+(u)F_{2,1}^+(u)\Big)\,.$$

Taking difference of two equalities in (C.134) and applying the projection $P_f^+$ to this difference one obtains

$$f(u,v)P_f^+\left(F_{4,2}^+(u)F_1(v)\right) = g(u,v)F_{4,1}^+(v) + \frac{1}{f(v\xi,u)}\,F_{2,1}^+(v)F_{4,2}^+(u)$$
$$+ \frac{g(u,v\xi)}{qf(v\xi,u)}\left(F_{2,1}^+(v)F_{3,1}^+(v) + F_{2,1}^+(v)^2F_{3,2}^+(u)\right),$$

which after setting $u = v$ implies

$$P_f^+\left(F_{4,2}^+(u)F_1(u)\right) = F_{4,1}^+(u).$$

Using latter equality and (C.132) one gets

$$P_f^+(F_3(u)F_2(u)F_1(u)) = P_f^+(F_{4,2}^+(u)F_1(u)) = F_{4,1}^+(u). \tag{C.135}$$

Equalities (C.132), (C.133) and (C.135) together with analogous relations for the currents $E_i(u)$ are base of the induction to prove Proposition 7.1.

# D  Proof of Proposition 7.1

We start with the proof of the first equality in (7.74). Assume that it is valid in the algebra $U_q^{n-1}(\tilde{\mathfrak{g}})$. It means that in order to prove first equality in (7.74) one has to prove it for the Gauss coordinates $F_{N,i}^+(u)$, $1 \leq i \leq N-1$ and $F_{j,1}^+(u)$, $2 \leq j \leq N$ using induction assumption that it is valid for all $F_{j,i}^+(u)$, $2 \leq i < j \leq N-1$.

To do this we consider RLL commutation relations (3.27) written in the form

$$f(u,v)f(v,u)\,(\mathbb{I}\otimes L^+(v))\cdot(L^-(u)\otimes\mathbb{I})$$
$$= R_{12}(u,v)\cdot(L^-(u)\otimes\mathbb{I})\cdot(\mathbb{I}\otimes L^+(v))\cdot R_{21}(v,u). \tag{D.136}$$

Consider $(i, i+1)$ matrix element in the first space and $(i+1, N)$ matrix element in the second space of the equality (D.136). After substitution in the resulting equality Gauss decomposition (4.37) one can multiply it from the right by $k_{i+1}^+(v)^{-1}$ and $k_i^-(u)^{-1}$ and normal order products of Gauss coordinates according to the ordering (4.41). Then one has to restrict resulting equality to subalgebra $U_f^+$ as it was described in the section 4.1, multiply it by $(u-v)^3$ and set $u = v$. Final equality takes the form

$$(u-v)\left(F_{N,i+1}^+(v)k_{i+1}^+(v)F_{i+1,i}^-(u)k_{i+1}^+(v)^{-1} + F_{N,i}^+(v)k_i^+(v)[E_{i,i+1}^+(v),F_{i+1,i}^-(u)]k_{i+1}^+(v)^{-1}\right)\Big|_{U_f^+}\Big|_{u=v} = 0.$$

Using the commutation relations $k_{i+1}^+(v)$ and $E_{i,i+1}^+(v)$ with $F_{i+1,i}^-(u)$ which are different for different $U_q(\tilde{\mathfrak{g}})$ when $i = n+1$ and taking into account that restriction to subalgebra $U_f^+$ coincides with the action of the projection $P_f^+$ onto subalgebra $U_f$ one obtains the equality in $U_f^+$

$$F_{N,i}^+(v) = P_f^+\left(F_{N,i+1}^+(v)\left(F_{i+1,i}^+(v) - F_{i+1,i}^-(v)\right)\right) = P_f^+\left(F_{N,i+1}^+(v)\,F_i(v)\right), \tag{D.137}$$

where $i < N-1$ for all $U_q(\tilde{\mathfrak{g}})$ except $U_q(D_n^{(1)})$ and

$$F_{N,i}^+(v) = P_f^+\left(F_{N,i+2}^+(v)\left(F_{i+2,i}^+(v) - F_{i+2,i}^-(v)\right)\right), \tag{D.138}$$

for the algebra $U_q(D_n^{(1)})$ at $i = n-1, n$. To obtain (D.138) one can use $(i, i+2)$ matrix element in the first space and $(i+2, N)$ matrix element in the second space of the equality (D.136) since $F_{n+1,n}^-(u) \equiv 0$ for the algebra $U_q(D_n^{(1)})$.

The statement of the Proposition is obviously valid for $i = N - 1$ since $\mathrm{F}^+_{N,N-1}(u) = P^+_f(F_{N-1}(u))$. Assume that it is valid for the Gauss coordinate $\mathrm{F}^+_{N,i+1}(u)$ with $i < N - 2$. Then equality (D.137) takes the form

$$\mathrm{F}^+_{N,i}(v) = P^+_f\Big(P^+_f\big(F_{N-1}(v)\cdots F_{i+1}(v)\big)\, F_i(v)\Big).$$

The statement of the Proposition 7.1 for the Gauss coordinate $\mathrm{F}^+_{N,i}(v)$ is proved by induction since projection $P^+_f$ possesses the property [3] that for $i < j$

$$P^+_f\big(F_{j-1}(u)\cdots F_i(u)\big) = F_{j-1}(u)\cdots F_i(u) + \dots , \tag{D.139}$$

where $\dots$ stands for the terms annihilated by the projection $P^+_f$.

Taking in (D.136) matrix entries $(1,2)$ in the first space and $(2,j)$ in the second space, using Gauss decompositions (4.37) for the matrix elements of L-operators, multiplying resulting equality by $(u-v)^3$ and by $k^-_1(u)^{-1}k^+_2(v)^{-1}$ from the right, normal ordering of the Gauss coordinates and restricting to subalgebra $U^+_f$ one obtains after setting $u = v$

$$\mathrm{F}^+_{j,1}(v) = P^+_f\Big(\mathrm{F}^+_{j,2}(v)\, F_1(v)\Big).$$

Using induction assumption for the Gauss coordinate $\mathrm{F}^+_{j,2}(v)$ and (D.139) we finish proof of the first equality in (7.74).

To prove the second equality in (7.74) we have to repeat all arguments as above for the transpose-inverse L-operators $\hat{\mathrm{L}}^\pm(u)$ using Gauss decomposition (4.40). Taking the corresponding matrix elements in (D.136) one can find that

$$P^-_f\Big(F_{j-1}(v)\, \tilde{\mathrm{F}}^-_{j-1,1}(v)\Big) = \tilde{\mathrm{F}}^-_{j,1}(v)$$

and

$$P^-_f\Big(F_{N-1}(v)\, \tilde{\mathrm{F}}^-_{N-1,i}(v)\Big) = \tilde{\mathrm{F}}^-_{N,i}(v).$$

By the property of projection $P^-_f$ [3]

$$P^-_f\Big(F_j(u)\cdots F_i(u)\Big) = F_j(u)\cdots F_i(u) + \dots ,$$

where $\dots$ stands for the terms which are annihilated by projection $P^-_f$ and induction assumption one can prove the second equality in the first line of (7.74). The second line in (7.74) can be proved similarly. $\qquad\square$

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
