# Peer review of "On the R-matrix realization of quantum loop algebras"

_SciPost Physics, doi:SciPost Phys. 12, 146 (2022)_

## Round 1 · Referee Report · Anonymous (Referee 1) · 2021-12-14

Report

This paper is about the representation of quantum group algebras using R-matrix and $RLL$ relation. Such algbras are known to have two presentations, in terms of an infinite set of Cartan generators gathered in the L-Matrix or in terms of a finite set of Chevaley generators. It is the purpose of this paper to provide the relation between the two presentations for most of the Lie algebras ($D^{(2)}_n$ is left for a future work). Starting from the Gauss decomposition of the L-matrix, the authors obtain the presentation of the q-deformed Lie algebras in terms of currents generalizing the known result for $U_q(SL_N)$ and more recent results of refs [7-8].

The paper gathers many results and proofs such as the description of the central elements of $U_q(G)$ in terms of L-matrix elements, the Gauss decomposition of the L-matrix, various embeddings of smaller rank algebras. As such it is a good review article and will be useful to people working in intergrable systems and interested in the quantum group representations of Lie algebras. I recommend its publication.

---

## Round 1 · Referee Report · Anonymous (Referee 2) · 2022-1-9

Report

The paper studies the R-matrix realizations of quantum loop algebras of types $A_{N-1}^{(1)}$, $B_{n}^{(1)}$, $C_{n}^{(1)}$, $D_{n}^{(1)}$, $A_{N-1}^{(2)}$ in a uniform way. Using the Gauss decomposition of the fundamental $L$-operator, the authors constructed the current generators of the corresponding quantum loop algebras and obtained the relations between the current generators in terms of generating series. The results for cases $A_{N-1}^{(1)}$, $B_{n}^{(1)}$, $C_{n}^{(1)}$, $D_{n}^{(1)}$ were discussed in references [6-8]. The interesting case for this paper is the type of $A_{N-1}^{(2)}$. I think it is good to show an isomorphism between these two realizations (physicists may be less interested in such statements).

The paper discussed - the properties of R-matrices,
- the central elements in the quantum groups in terms of $L$-operator, - Gauss decompositions and current generators, - embedding theorem from smaller algebras into bigger algebras, - commutator relations for current generators, - certain projections which could possibly used to construct off-shell Bethe vectors.

I think the paper is interesting and should also be important for studying quantum integrable systems like the associated XXZ models and representations of quantum loop algebras such as R-matrix constructions of finite-dimensional irreducible modules (fusion procedure). Therefore, I recommend publishing it.

Requested changes

Articles are missing or used incorrectly many times. There are also typos related to singular and plural nouns. Let me list a few of them.

  • Page 0 Paragraph 3 Sentence 1: Let $q\in\mathbb C$ be arbitrary complex $\rightarrow$ Let $q\in\mathbb C$ be an arbitrary complex
  • Page 1 Sec. 2 Sentence 2: Let $e_{ij}$ be a $N\times N$ $\rightarrow$ Let $e_{ij}$ be an $N\times N$
  • Page 3 the line above the displayed formulas defining $P$ and $Q$: onto one-dimensional subspace $\rightarrow$ onto a one-dimensional subspace
  • Page 4 bottom: has simple pole at $\rightarrow$ has a simple pole at
  • Page 8 Prop. 3.2: means that product of the matrices $\rightarrow$ means that products of the matrices
  • Page 9 proof of Prop. 3.2: chain of equalities $\rightarrow$ a chain of equalities
  • Page 9 bottom: given by (3.12) satisfies $\rightarrow$ given by (3.12) satisfy
  • Page 13 above eq. (5.5): and (5.4) implies that $\rightarrow$ and (5.4) imply that
  • Page 16 after the first displayed equation: which is consequence of $\rightarrow$ which is a consequence of
  • Page 16 beginning of Sec. 6: New realization of $\rightarrow$ A new realization of
  • Pages 19 – 22: Many “which includes” should be changed to “which include” since “relations” are used before “which”
  • Page 22 Sec 7 Paragraph 1 Sentence 2: Rigorous definitions of these projections depends on $\rightarrow$ Rigorous definitions of these projections depend on

Here are some other comments/suggestions.

  • Since the title is about quantum loop algebras, I suggest add "which we call the quantum loop algebra" after "$U_q(\widetilde {\mathfrak g})$ [2]" in Paragraph 3 Sentence 2 of the introduction
  • Page 4 the last sentence about Twist symmetry of R-matrix: This sentence does not seem to be fine. Please check it
  • Page 6 footnote: write simple $\rightarrow$ write simply
  • Page 10 the end of the first sentence in Sec 4: "$U_q(\widetilde {\mathfrak g})$ quantum affine algebras $U_q(\widehat {\mathfrak g})$" does not seem to be fine. Please check it
  • Page 11 Sec 4.1 Paragraph 2 Sentence 1: Please add ", respectively"
  • Page 12 Line 3 of the last paragraph of Sec 4.1: different type Borel subalgebras $\rightarrow$ different types of Borel subalgebras
  • Page 13 after Lem 5.2: them invertible operator $\mathrm{L}_{1,1}(v)$ $\rightarrow$ the invertible operator $\mathrm{L}_{1,1}(v)$?
  • Page 14 Eq. (5.8): Dot at the end of (5.8) is missing.
  • Page 21 before "and there are additional Serre relations": The dot should be changed to ","
  • Page 22 at the end of Sec 6.5 after "as in the case of $U_q(C_n^{(1)})$ (6.6)": The dot should be changed to ","

  • validity: -
  • significance: -
  • originality: -
  • clarity: -
  • formatting: -
  • grammar: -

Author:  Stanislav Pakuliak  on 2022-01-12  [id 2091]

(in reply to Report 2 on 2022-01-09)
Category:
remark
correction

We thank referee for the report on our paper and accept all requested changes he proposed to include in the text of the paper. We will prepare a new version of the manuscript according to these changes and send it to the SciPost when the paper will be accepted for publication.

---

## Round 3 · Referee Report · Anonymous (Referee 2) · 2022-2-20

Report

I would like to thank the authors for taking care of my suggestions. I have no further suggestions/comments. The paper deserves to be published.

---

## Round 3 · List of Changes

We include in the manuscript all corrections sent by Referee N2

Since the publication of the paper was rather delayed we changed the funding information and affiliations of one of the author

---

## Editorial Decision

published